# Predicting Functional Brain Connectivity with Context-Aware Deep Neural Networks

**Alexander Ratzan**[1,†]**, Sidharth Goel**[1]**, Junhao Wen**[3]**, Christos Davatzikos**[4]**, Erdem Varol**[1,2,†]

[1]Department of Computer Science, New York University
[2]Neuroscience Institute, Grossman School of Medicine, New York University
[3]Department of Radiology, Columbia University
[4]Department of Radiology, University of Pennsylvania

[†]*Correspondence:* `asr655@nyu.edu, ev2240@nyu.edu`

## Abstract

Spatial location and molecular interactions have long been linked to the connectivity patterns of neural circuits. Yet, at the macroscale of human brain networks, the interplay between spatial position, gene expression, and connectivity remains incompletely understood. Recent efforts to map the human transcriptome and connectome have yielded spatially resolved brain atlases, however modeling the relationship between high-dimensional transcriptomic data and connectivity while accounting for inherent spatial confounds presents a significant challenge. In this paper, we present the first deep learning approaches for predicting whole-brain functional connectivity from gene expression and regional spatial coordinates, including our proposed *Spatiomolecular Transformer* (SMT). SMT explicitly models biological context by tokenizing genes based on their transcription start site (TSS) order to capture multi-scale genomic organization, and incorporating regional 3D spatial location via a dedicated context `[CLS]` token within its multi-head self-attention mechanism. We rigorously benchmark context-aware neural networks, including SMT and a single-gene resolution Multilayer-Perceptron (MLP), to established rules-based and bilinear methods. Crucially, to ensure that learned relationships in any model are not mere artifacts of spatial proximity, we introduce novel spatiomolecular null maps preserving key transcriptomic autocorrelation structure. Context-aware neural networks outperform linear methods, significantly exceed our stringent null map estimates, and generalize across diverse connectomic datasets and parcellation resolutions. Together, these findings demonstrate a strong, predictable link between the spatial distributions of gene expression and functional brain network architecture, and establish a rigorously validated deep learning framework for decoding this relationship. Code to reproduce our results is available at: `github.com/neuroinfolab/GeneEx2Conn`.

## 1 Introduction

Throughout development and into adulthood, the coordinated expression of thousands of genes shapes the molecular and structural scaffold of the brain [1, 2]. Functional brain networks emerge from this scaffold through the synchronous activity of multiscale neural components [3] . This emergent organization is central to cognition, supporting processes such as vision, language, and memory with disruptions to these networks linked to a range of neuropsychiatric and neurodegenerative disorders [4]. Understanding the spatiomolecular landscape that gives rise to functional brain networks is a critical step towards uncovering the genetic basis of brain function and dysfunction. Thus, we set

39th Conference on Neural Information Processing Systems (NeurIPS 2025).

out to address the fundamental hypothesis that the biological richness of gene expression is highly predictive of functional connectivity [5–8].

Recent advances in brain-wide gene expression atlases and large neuroimaging datasets have made it possible to connect spatial variations in gene expression with the organization of functional brain networks [9, 10]. However, the path to robust predictive models has several key obstacles. First, the scale of transcriptomic data, involving thousands of genes with complex co-expression patterns, demands models with high expressive capacity. Conventional approaches, often favoring bilinear factor models for their perceived explainability, may not adequately capture inherent non-linearities [11, 2, 12–15, 7]. Second, a major confound in neurogenomic studies is spatial autocorrelation: nearby brain regions often share similar gene expression and connectivity patterns due to shared developmental trajectories, vascularization, or signal bleed, potentially inflating statistical associations if not rigorously controlled [16, 17]. Third, obtaining directly paired brain-wide gene expression and *in-vivo* functional connectivity data from the same human individual is challenging due to postmortem collection of gene expression, motivating the use of carefully aggregated population atlases from multiple data sources. Lastly, effectively integrating biological context—such as the genomic organization of genes or the precise 3D spatial embedding of brain regions—into predictive models in a meaningful way beyond simple feature engineering, remains an open question.

To address these challenges, we introduce context-aware neural network architectures including the *Spatiomolecular Transformer* (SMT) designed specifically for predicting functional connectivity from regional gene expression and spatial coordinates (Figure 1) evaluated through a rigorous experimental setup. Inspired by single-cell approaches such as spaCI [18] and scBERT [19] that use multimodal contextual information to improve performance on downstream tasks, SMT derives brain region embeddings from gene expression sequences tokenized based on reference genome position (transcription start site, TSS) and incorporates regional spatial location information via a dedicated [CLS] token. Critically, SMT utilizes Attention with Linear Biases (ALiBi) [20] in its bidirectional multi-head self-attention mechanism to encourage hierarchical transcriptomic representations of the input sequence. Furthermore, to counteract the limited sample size of human gene expression data [21], we leverage the large corpus of individual connectomes in our fMRI datasets [22–24] through a distributional target-side augmentation approach during training [25]. All methods are evaluated under brain-wide and spatially constrained train-test splits alongside a novel spatiomolecular null mapping technique to assess if performance is inflated by spatial autocorrelation.

As such, we claim the following **main contributions** in this paper:

- We introduce the first deep learning approaches for predicting functional connectivity from regional gene expression in humans, featuring an attention-based architecture (SMT) with biologically-informed components for targeted hypothesis testing.

- We introduce and validate a novel spatiomolecular null mapping evaluation technique that generates surrogate gene expression maps preserving not only spatial autocorrelation but also key transcriptomic correlation structures, providing a highly stringent benchmark for assessing genuine predictive signal.

- Context-aware neural networks, including SMT and a single-gene resolution Multilayer-Perceptron, achieve significant performance gains in predicting functional connectivity as compared to rules-based and bilinear methods. Crucially, this performance significantly exceeds that of our rigorous null models and is shown to generalize across multiple connectomic datasets (UK Biobank [22], Human Connectome Project [23], MPI-LEMON [24]) and parcellation resolutions.

Our findings highlight that the complex, multimodal architecture of functional brain networks can be predicted from spatiomolecular features using context-aware neural networks, well beyond what chance or spatial proximity would dictate. Context-aware neural networks thus offer a robust framework for multimodal modeling of brain function in health and disease.

## 2 Preliminaries

RNA-sequencing (RNA-seq) provides a snapshot of transcriptomic activity across single-cell or bulk tissue samples, typically summarized in a gene expression matrix as in Figure 1. Gene expression has recently been integrated with noninvasive neuroimaging to uncover molecular correlates of brain

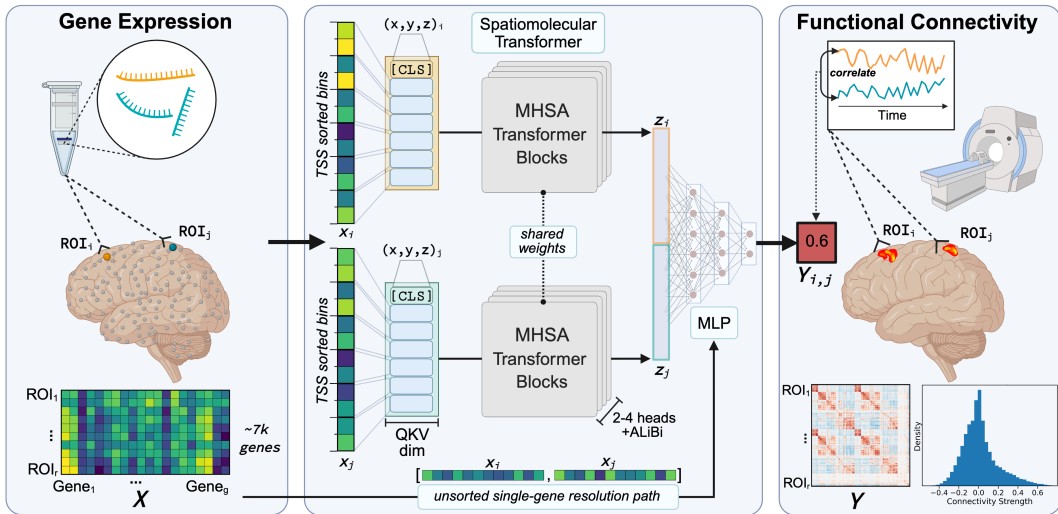

Figure 1: **Spatiomolecular context-aware deep modeling for connectivity prediction.** Gene expression from the Allen Human Brain Atlas is sampled across the entire brain. The Spatiomolecular Transformer (SMT) uses a reference genome based tokenization strategy and multi-head self attention (MHSA) to encode gene expression profiles for region-of-interest (ROI) $i$ and $j$. A learnable [CLS] token, initialized based on 3D coordinates of sampled ROIs, is optionally passed as a token to the SMT. Concatenated embeddings are used to decode population average connectivity. Alternatively, a fully connected Multilayer-Perceptron can be trained using concatenated gene expression vectors.

organization [26, 2, 27, 7]. Resting-state functional MRI (fMRI), collected during task-free windows, characterizes intrinsic brain activity by computing functional connectivity as the Pearson correlation of BOLD (blood-oxygen-level-dependent) signals between brain regions [4]. Letting $X \in \mathbb{R}^{r \times g}$ represent a population-average transcriptome and $Y \in \mathbb{R}^{r \times r}$ a symmetric connectome, the goal is to learn a predictive mapping $f \colon X \to Y$, or equivalently $f(x_i, x_j) = Y_{i,j}$, testing the fundamental hypothesis that the biological richness of gene expression is predictive of whole brain functional connectivity at the atlas-level. Our study leverages high resolution brain maps carefully aggregated from population datasets to learn a predictive mapping between the transcriptome and connectome.

## 2.1 Datasets

**Allen Human Brain Atlas (AHBA).** With approximately 500 spatial locations sampled in each hemisphere across the cortex, subcortex, and cerebellum, the Allen Human Brain Atlas is the most spatially-resolved human gene expression dataset to date. Microarray data was collected from six neurotypical donors (mean age=42.5, sex ratio=5:1 (M:F)) using whole-genome Agilent microarrays [7]. Data is processed using recommendations from the *abagen* package ensuring stable measurements through a differential stability threshold, filtering genes based on background noise and consistent inter-regional coexpression patterns across donors. Since we are relating gene expression across populations, we use the most stringent differential stability threshold retaining a final set of 7,380 genes averaged per region across donors [3, 21]. Detailed processing steps are outlined in Section A.1. Raw data is available at https://portal.brain-map.org/.

**UK Biobank (UKBB).** The primary connectivity dataset in our study is a subset of $n = 1814$ healthy participants from the UK Biobank (mean age=63.3 years; age range=45-82; 55% female) with available resting-state functional MRI (rs-fMRI) [22]. Scanning protocols and preprocessing steps are outlined in Section A.1. Functional timeseries were extracted using the 7-network Schaefer 400 parcel cortical atlas [28], extended to 456 regions with the inclusion of the Tian subcortical atlas [29] (S456 parcellation). Pairwise Pearson correlation of parcel-level BOLD signals was used to compute functional connectivity matrices. The population averaged 456 region UKBB dataset yields 103,740 edge-level targets and corresponding region-pair inputs from AHBA for model optimization (see Figure 2A).

Alongside UKBB, the Human-Connectome-Project Young-Adult (HCP-YA) [23] and Max Planck Institute Leipzig Mind-Brain-Body Dataset (MPI-LEMON) [24] are used as validation connectomic datasets. Despite substantial age shift between these datasets and UKBB, we observe a consistent backbone connectivity structure in the population average functional connectomes across datasets (Figure 4). Processing details for HCP-YA and MPI-LEMON are outlined in Section A.1.

## 2.2 Baselines and related works

To rigorously evaluate our context-aware deep neural networks, we compare them against three classes of established models from the neuroimaging and connectomics literature.

**Rules-based methods.** We first include simple, interpretable models that test foundational hypotheses of brain organization. These include an *exponential decay model* [17, 30], assuming connectivity strength decreases with Euclidean distance. A related model uses a *Gaussian kernel*, $Y_{ij} = \exp(-d_{ij}^2/2\sigma^2)$, which decays symmetrically with squared distance. We also include a *Correlated Gene Expression (CGE)* model, computed by correlating PCA-reduced gene expression profiles between pairs of regions, effectively predicting connectivity based on molecular similarity [4, 31]. While interpretable, these methods are constrained by their fixed forms and feature sets.

**Connectome Model.** Second, we implement learned bilinear models prominent in connectomics [11, 12]. The *Connectome Model (CM)*, introduced by Kovács et al. [11], instead formulates synaptic connectivity prediction as a learned bilinear regression problem, modeling the connectome as $Y = XOX^\top$, where $X$ is a single neuron gene expression matrix and $O$ is an unknown gene-gene interaction matrix. Qiao [12] extends the Connectome Model by introducing a *Bilinear Low-rank* decomposition of $O$. Implementation details for the Connectome Model and its low-rank counterpart can be found in Section A.3.

**Partial Least Squares Regression.** Finally, we include *Partial Least Squares (PLS) regression*, the predominant multivariate method in imaging transcriptomics for linking multiple data modalities [5, 7, 8, 2, 3, 9]. PLS identifies latent variables that maximize the covariance between gene expression and connectivity profiles. To adapt PLS for our edge-wise prediction task, we reformulate it into an encoder-decoder model: we use its learned shared projections to create region-level embeddings and then predict connection strength via a bilinear decoder. Details can be found in Section A.3.

This comprehensive suite of baselines allows us to benchmark the performance gains offered by our more expressive, non-linear architectures.

## 3 Methods

**Spatiomolecular Transformer (SMT).** Transformers have been adapted for numerous biological tasks due to the sequential nature of gene expression data [32] with notable advances in single-cell perturbation modeling, disease classification, and cell type annotation [19, 33, 34]. Similar methods remain largely unexplored for understanding the spatiomolecular foundations of brain connectivity. Motivated by successes in single-cell modeling, we formulate a transformer-based architecture for the transcriptome-connectome prediction task. Despite the scarcity of spatially-resolved human brain gene expression data, we leverage the abundant amount of connectomics data to train a context-aware transformer end-to-end.

Here, we adopt a *BERT-style multi-head self-attention (MHSA)* based transformer architecture [35], requiring tokenization of the gene expression input. Given the dimensionality of the gene space, $g = 7380$, we partition the gene expression vector of each region $x_i \in \mathbb{R}^g$ into contiguous non-overlapping bins of $k$ genes sorted by transcription start site (TSS) on the human reference genome (see Section A.2 for details). Setting $k = 60$ genes yields $\ell = g/k = 123$ tokens per region, each represented as a length-$k$ scalar vector. These are projected into an embedding space via a learned linear map, forming input $X \in \mathbb{R}^{\ell \times d}$, with embedding dimension $d = 128$. This *value projection* encoding strategy is effectively leveraged by models like TOSICA [33] and scBERT [19].

Each token embedding sequence is passed through up to 4 layers of MHSA with 2-4 attention heads, where the input $X$ is linearly projected into queries, keys, and values as $Q = XW_Q$, $K = XW_K$, and $V = XW_V$, with $W_Q, W_K, W_V \in \mathbb{R}^{d \times d_h}$ and $d_h = d/h$. To incorporate positional priors

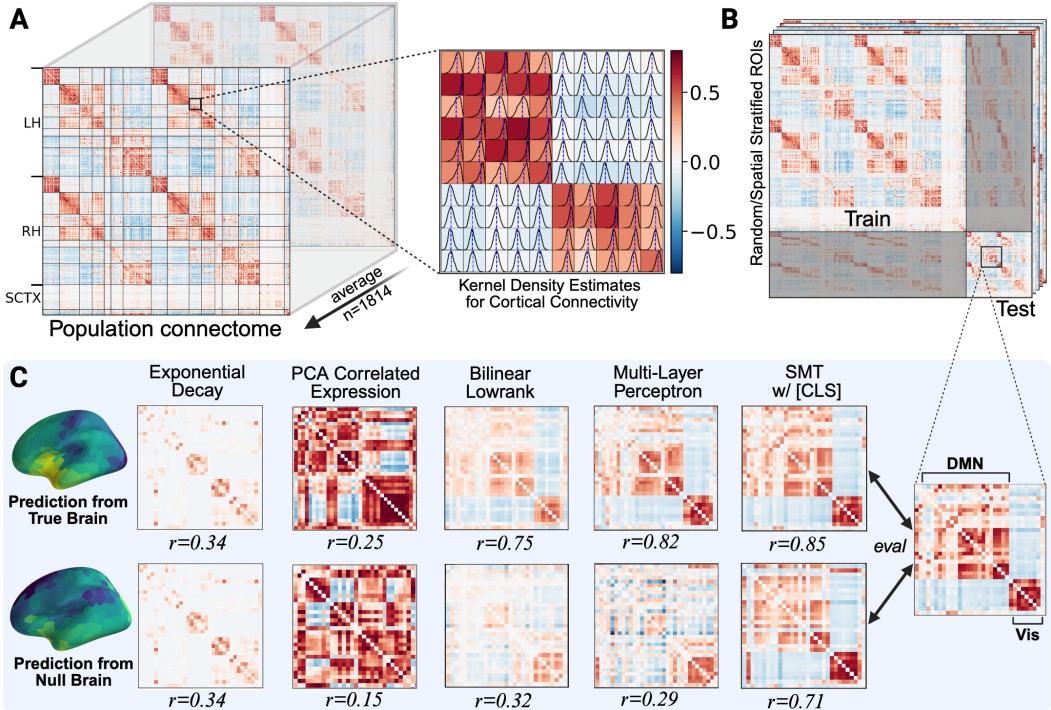

Figure 2: **Population connectome reconstruction.** **[A]** Average population connectome, $Y$, with delineated subnetworks of Schaefer 7-network 400 region parcellation [28] (LH=left hemisphere, RH=right hemisphere, SCTX=subcortex). KDEs are plotted for a subset of cortical regions with vertical blue lines representing the mean. Connectivity values are probabilistically sampled during training from the underlying population (e.g. UK Biobank, $n$=1814) under our target augmentation protocol. **[B]** Connectomes are stratified into 10 four-fold train-test splits based on a random or spatial split strategy. Edges connecting train and test sets are omitted. **[C]** For each split, models are trained using both the true brain and a spatiomolecular null brain. Reconstructed $Y_{test}$ for select models are displayed with comparison to null reconstructions for an example random split.

without learned embeddings, we add *ALiBi (Attention with Linear Biases) slopes* [20] to the attention logits, implemented using FlashAttention [36] to enable scaling to hundreds of tokens (and thus thousands of genes) per region. The fixed head-specific penalties bias each head toward different ranges of token interactions:

$$\text{Attention}(Q, K, V) = \text{softmax}\left(\frac{QK^{\top} + B}{\sqrt{d_h}}\right) V,$$

where $B$ encodes relative-position biases. Steep slopes prioritize local interactions while shallow slopes attend globally. This inductive bias allows the model to co-embed genes at multiple genomic scales—motivated by the bidirectional organization of functional gene groups within and across chromosomes [37, 38]. Given that genomic interactions may be coarse at the macroscale of whole-brain networks, ALiBi over $k$ gene tokens encodes genomic proximity as a soft prior, while *value projection*–based tokenization preserves single-gene–level signal relevant to connectivity prediction. Furthermore, *chromosomally organized tokens* can be compared post-hoc with known functional gene groups or Genome-Wide Association Studies for deeper biological insight [39].

Following transformer layers, the final output is flattened and linearly projected to form a region embedding $z_i \in \mathbb{R}^p$, where $p = \ell \cdot d_{\text{out}}$ and $d_{\text{out}} \leq 10$. For a given region pair, the concatenated embedding $[z_i \,\|\, z_j]$ is passed through a 2 or 3-layer MLP decoder to predict connectivity $\hat{Y}_{ij}$.

**Single-gene resolution Multilayer Perceptron (MLP).**    In parallel, we implement a fully connected multilayer perceptron (MLP) with up to four hidden layers. The input to the model is the concatenated single-gene resolution gene expression vector from a pair of regions, of dimensionality $2 \times 7380$ genes. Architecture and optimization details for SMT and MLP can be found in Section A.3.

**Incorporating spatial context awareness.** Spatial position is another form of biological context that can be introduced to our non-linear models, with well documented links to connectivity strength [27, 7, 40, 17]. Here, we extend the input sequence with spatial information by concatenation of MNI (Montreal Neurological Institute) 3D coordinates for the MLP, and for the SMT through a dedicated `[CLS]` token initialized by the coordinates of each brain region. This token is linearly projected into the same embedding dimension as the gene expression tokens and participates fully in multi-head self-attention as in Temporal Fusion Transformer [41]. Unlike BERT, the full sequence including the `[CLS]` token is used as the output representation of the sequence, instead of using the `[CLS]` token exclusively. For both the MLP and SMT, we interpret spatial coordinates as context carriers that condition the encoding of molecular information based on anatomical location in the brain.

**Distributional target-side augmentation.** To improve model robustness and generalization beyond the population-averaged connectome, we introduce a distributional target-side augmentation strategy. Rather than training exclusively on the mean connectivity matrix $Y \in \mathbb{R}^{r \times r}$, we leverage the full distribution of individual-level connectomes $Y' \in \mathbb{R}^{r \times r \times n}$. Inspired by curriculum learning principles [42] and imbalanced data sampling [43], we expose the model to a subset of subject-specific targets at different stages of training. By replacing static supervision with a set of $n \cdot r^2$ dynamic targets, this strategy injects meaningful variance into the loss landscape without altering evaluation, which remains based on mean squared error against $Y$. We use a decaying schedule augmentation strategy for the SMT model across all datasets in an effort to improve test-set generalization. Algorithmic details and scheduling strategy experiments are outlined in Algorithm 2 and Section A.5.

## 4 Evaluation

**Train-test split.** To rigorously evaluate model performance, we employ two train-test split protocols: *random* and *spatial*. Each protocol is repeated across 10 rounds of 4-fold cross-validation, resulting in 40 total splits per model. For both settings, model specific hyperparameters are tuned using a nested cross-validation procedure on mean-squared error described in Section A.3.

For each random split, 75% of brain regions are used for training and 25% for testing, with models trained on the $\binom{r_{\text{train}}}{2}$ training-region edges and evaluated on the disjoint $\binom{r_{\text{test}}}{2}$ test edges (Figure 2B). This split tests a model's ability to generalize across regions that may be sampled from the same underlying distribution, allowing it to leverage the full diversity of functional connections across the brain. To more stringently test spatially invariant generalization, we implement a *spatial split* protocol adapted from Hansen et al. [1]. For each fold, we select a source region and define the test set as the 25% of brain regions that are spatially closest based on Euclidean distance in Montreal Neurological Institute (MNI) space. The remaining 75% of regions comprise the training set. This is repeated across four folds, ensuring full brain coverage. This more challenging protocol tests a model's ability to generalize to regions that are both unseen and spatially distinct from those in training. Both splits are visualized in Figure 11.

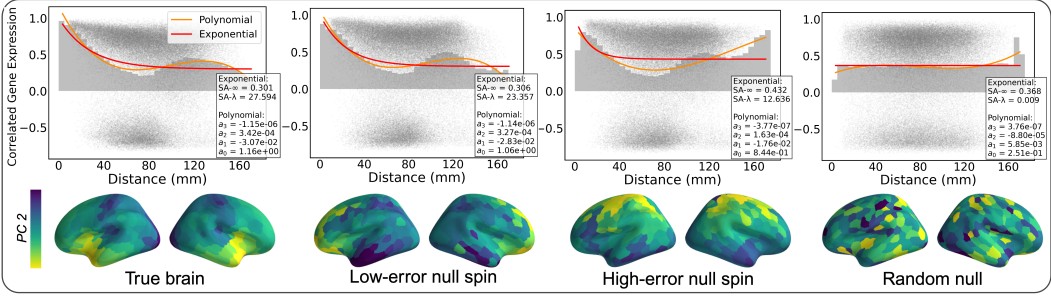

Figure 3: **Spatiomolecular null brain maps.** Example null brain maps generated from Algorithm 1. Parcel color represents the second principal component score of gene expression matrix, $X$, at each ROI. Polynomial and exponential decay curves are fit using 5mm bins to model correlated gene-expression vs. distance. Curve parameters are compared between true and null brains to measure how well each null brain map preserves realistic transcriptomic autocorrelation.

**Spatiomolecular null mapping.**    Our train-test split setup evaluates generalization to unseen brain regions by using common splits across all models, but it does not fully account for a critical confound in neurogenomic data - spatial autocorrelation. Adjacent brain regions often share similar gene expression and connectivity patterns, which can inflate statistical associations [16]. To disentangle these effects, we implement spatially constrained null models that preserve realistic autocorrelation structure between brain regions while shuffling expression-to-region assignments.

We adopt the procedure of Váša et al. [44], performing 10,000 spatial rotations, or 'spins', of the gene expression matrix and selecting null brains that best match the empirical correlated gene expression (CGE) vs. distance curve (for details see A.4 and Algorithm 1). Traditional spin tests preserve geometric features like distance decay but do not account for transcriptomic structure [45]; our CGE-matched nulls address both. This approach aligns with recent trends using null brain map rejection sampling to minimize false positive inflation [46]. Figure 3 shows the second principal component of the gene expression matrix projected onto the cortical surface for the true brain and three possible null brain maps. The low-error null spin highlights the global smoothness preserved by a null brain matching the true CGE vs. distance relationship. Conversely, a naive randomly shuffled brain map completely destroys the CGE vs. distance trend.

For each four-fold train-test split, models are retrained and evaluated using one of 10 lowest error spatially shuffled gene expression inputs while keeping the original connectome target the same. This enables us to test whether models trained on true data are capturing genuine biological transcriptome–connectome patterns rather than relying on mere spatial autocorrelation or genetic similarity.

**Performance metrics.**    True and null model performance is evaluated using a range of global metrics including Pearson correlation, $R^2$, mean squared error (MSE), and geodesic distance computed on the Riemannian manifold of symmetric positive semidefinite correlation matrices [47]. Geodesic distance serves as a complementary measure of similarity in global correlation structure, potentially capturing topological properties missed by elementwise metrics. We also report stratified correlations: (i) by connection distance using uniform bins up to 180 mm, (ii) by subnetwork, based on whether a test connection is, for example, intra- or inter-Visual (iii) by edge strength where $|r| > 0.3$ . Tables 1, 2, and Figure  5 present a subset of these metrics; the full 32 metric sets for all experiments are available in our repository, `github.com/neuroinfolab/GeneEx2Conn/notebooks/NeurIPS`.

## 5    Results

**Overall performance.**    A primary finding of our study is that non-linear architectures as a class consistently outperform linear and rules-based methods. Both the SMT and MLP, trained only on gene expression, achieve the highest global performance for both random and spatial train-test splits (Table 1). Performance gains are especially pronounced under the stringent spatial split, with improvement gains exceeding 0.08 Pearson-$r$ for long-range and strong positive connections (>0.3) as compared to the best linear method (Table 2).

Table 1: Global metric test-set performance on UKBB dataset for random and spatial splits (mean±SD over 10 four-fold splits. Best metric-wise performance per section is **bolded**)

| | **Random Split** | | | **Spatial Split** | | |
|---|---|---|---|---|---|---|
| **Model** | Pearson-$r$ (True/Null) | MSE | Geodesic | Pearson-$r$ (True/Null) | MSE | Geodesic |
| CGE (PCA) | .23±.03 / .17±.04 | .158±.011 | **15.1**±0.8 | .31±.09 / .23±.07 | .191±.042 | **13.5**±2.4 |
| Dist. Gauss. Kernel | .30±.02 / - - ± - - | .031±.002 | 18.3±1.7 | .41±.05 / - - ± - - | .039±.009 | 14.4±1.9 |
| Dist. Exp. Decay | **.35**±.03 / - - ± - - | **.030**±.003 | 15.7±0.7 | **.44**±.06 / - - ± - - | **.036**±.009 | 14.0±1.3 |
| Bilinear CM (PCA) | .51±.04 / .32±.06 | .025±.003 | 15.3±0.6 | .42±.15 / .25±.13 | .080±.110 | 16.3±2.1 |
| Bilinear CM | .62±.10 / .21±.08 | .030±.012 | 16.1±1.4 | .44±.25 / .15±.12 | .076±.077 | 16.8±2.3 |
| Bilinear CM (PLS) | .72±.08 / .33±.08 | .016±.004 | 12.0±1.9 | .58±.17 / .26±.12 | .029±.010 | 14.5±4.1 |
| Bilinear Low-rank | .77±.03 / .42±.07 | .014±.001 | 11.2±1.1 | .66±.13 / .34±.13 | .025±.009 | 12.9±2.2 |
| MLP | .78±.03 / .29±.09 | .014±.001 | 10.3±1.1 | **.72**±.13 / .26±.12 | .023±.012 | **10.5**±1.9 |
| SMT | **.79**±.03 / .34±.09 | **.013**±.001 | **10.2**±1.0 | **.72**±.13 / .30±.12 | **.022**±.010 | 10.8±1.7 |
| MLP w/ coords | .83±.03 / .48±.09 | .011±.001 | **9.6**±0.8 | **.74**±.13 / .42±.11 | **.021**±.011 | **10.3**±2.1 |
| SMT w/ [CLS] | **.84**±.02 / .73±.05 | **.010**±.001 | 9.8±1.0 | .70±.12 / .49±.13 | .025±.012 | 11.1±2.0 |

*Null reflects model performance when trained on spatiomolecularly shuffled brains. Models sectioned by rules-based, learning-based with gene expression, learning-based with gene expression and spatial information.*

Table 2: Stratified metric test-set performance on UKBB dataset for random and spatial splits (mean±SD over 10 four-fold splits. Best metric-wise performance per section is **bolded**)

| | Random Split | | | | Spatial Split | | | |
|---|---|---|---|---|---|---|---|---|
| Model | Short-$r$ | Long-$r$ | Inter-hemi-$r$ | Strong-pos-$r$ | Short-$r$ | Long-$r$ | Inter-hemi-$r$ | Strong-pos-$r$ |
| CGE (PCA) | .25±.04 | **.19**±.08 | .20±.04 | .31±.05 | .30±.08 | **.07**±.26 | .31±.12 | .30±.08 |
| Dist. Gauss. Kernel | .40±.04 | .07±.05 | .17±.03 | .32±.05 | .43±.06 | .01±.08 | .29±.10 | .35±.09 |
| Dist. Exp. Decay | **.41**±.04 | .10±.07 | **.24**±.04 | **.34**±.05 | **.45**±.06 | -.01±.13 | **.37**±.13 | **.36**±.09 |
| Bilinear CM (PCA) | .57±.05 | .35±.07 | .50±.04 | .31±.08 | .42±.17 | .20±.24 | .39±.18 | .22±.12 |
| Bilinear CM | .63±.12 | .55±.09 | .58±.11 | .22±.10 | .43±.27 | .25±.20 | .27±.24 | .15±.19 |
| Bilinear CM (PLS) | .75±.08 | .63±.11 | .72±.08 | .37±.10 | .60±.18 | .38±.25 | .52±.19 | .32±.10 |
| Bilinear Low-rank | .80±.03 | **.70**±.06 | .77±.03 | .40±.07 | .67±.15 | .40±.28 | .57±.16 | .35±.10 |
| MLP | **.82**±.03 | .67±.06 | .77±.03 | **.48**±.08 | **.73**±.14 | **.50**±.26 | **.62**±.17 | **.43**±.11 |
| SMT | **.82**±.03 | .69±.07 | **.78**±.03 | **.48**±.07 | .72±.15 | .49±.27 | **.62**±.17 | .40±.11 |
| MLP w/ coords | .85±.03 | .75±.06 | **.84**±.03 | .53±.07 | **.74**±.15 | **.58**±.22 | **.66**±.16 | **.46**±.12 |
| SMT w/ [CLS] | **.86**±.02 | **.78**±.06 | **.84**±.02 | **.55**±.06 | .70±.13 | .53±.26 | .62±.20 | .41±.13 |

Figure 5 quantifies non-linear gains across functional subnetworks, with the strongest improvements observed in unimodal cortex (e.g., Somatomotor, Visual). In complement, Figure 2 shows that linear models underperform in reconstructing distributed, strong positive connections within the Visual system and Default Mode Network. These patterns suggest that non-linear architectures are better equipped to capture the inherent complexity of such connections. Notably, the low-rank bilinear model from Qiao [12] performs competitively and offers greater robustness than the standard Connectome Model, regardless of single-gene or PCA-reduced inputs, suggesting it may be optimal to learn embeddings in a supervised fashion (e.g. PLS, low-rank projection) for linear methods.

**Performance above spatiomolecular null shuffle.**    Crucially, both linear and non-linear models significantly outperform their respective spatiomolecular null evaluations, indicating that learned representations capture molecular interactions beyond spatial autocorrelation and gene co-expression. Null performance varies moderately across models, generally falling in the range of rules-based methods ($r = 0.3$-$0.4$), which explicitly model properties of spatial and molecular similarity.

**Spatial context effects.**    A performance boost is observed when spatial context is introduced to non-linear methods in the random split setting. This gain is less pronounced under the spatial split, where the SMT with [CLS] token exhibits stronger overfitting effects than the MLP. This is accompanied by high null performance ($r = 0.73$) highlighting that spatial coordinates alone are powerful predictors of connectivity, a signal the SMT with [CLS] architecture effectively leverages (Figure 13B). However, our stringent spatial split evaluation decisively shows that the model learns more than just this spatial signal. In that setting, the null model performance drops substantially while the true model's performance remains high, doubling the performance gap over the null and confirming the capture of genuine, spatially generalizable molecular information. Future architectural modifications could consider gating or regularization strategies to optimally leverage spatial context.

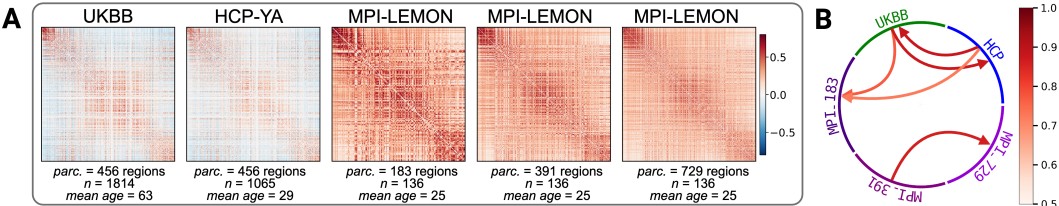

Figure 4: **Cross-dataset population connectome reconstruction. [A]** Multi-resolution population average connectomes sorted by posterior-anterior axis. **[B]** Chord plot representing Pearson-$r$ reconstruction when training SMT w/ [CLS] models on a full source dataset and predicting on a full target dataset. Cross-dataset arrows are visualized when exceeding the null, and within-dataset if a coarser resolution generalizes to a higher resolution (e.g. 391→729). See Section 12 for experimental details on cross-dataset generalization experiments.

**Multi-dataset replication.** SMT results are replicated using the HCP-YA and MPI-LEMON datasets by repeating the same random and spatial split protocols used in Table 1. Results are shown in Tables 5–8. Performance trends are consistent across datasets and parcellation resolutions, with all models exceeding their respective spatiomolecular null baselines. The SMT tends to produce more accurate predictions at higher resolutions, which may be a byproduct of increased pairwise training data at higher resolutions. Overall performance is somewhat attenuated in these datasets, particularly for MPI-LEMON, likely due to differences in preprocessing rather than biological signal—supported by similar cohort demographics between the validation datasets but differing ranges in connectivity strength.

**Cross-dataset generalization.** To test whether transcriptome-connectome mappings generalize across datasets, we evaluate whether a model trained on one dataset can predict another dataset's connectome above a spatiomolecular null brain map. For each connectomic dataset, we train the SMT with `[CLS]` using both a true and null brain at its native resolution. We then assess, for example, whether an SMT model trained on UKBB can predict the MPI-LEMON 183 connectome better than a null model trained on spatially shuffled gene expression in the native MPI-LEMON 183 resolution. This approach isolates generalizable cross-dataset transcriptomic patterns beyond spatial autocorrelation and the influence of spatial coordinates. As shown in Figure 4, SMT with `[CLS]` models trained on UKBB and HCP significantly outperform nulls when predicting MPI-LEMON 183 ($r = 0.78$, $r = 0.73$ respectively), and MPI-LEMON 391 generalizes strongly to the finer MPI-729 resolution ($r = 0.86$). Full experimental details are provided in Section A.5.

**Embedding evaluation.** The SMT's dedicated encoder enables direct visualization of learned region-level embeddings across the brain (Figure 5). Each region of interest (ROI) is represented by an embedding matrix of dimensionality $123 \times 10$ for 60 gene bins, obtained from a model trained on the full UKBB dataset using default hyperparameters. Evaluating embeddings by UMAP reveals a structured representation of the Visual system by the SMT, which, unlike the bifurcated organization observed in raw gene expression UMAP, forms a coherent transcriptomic subspace clustered by strong functional connectivity patterns. This suggests that the SMT captures a connectivity-relevant axis of transcriptomic variation that may be distinct from purely transcriptomic gradients. Other systems, such as the Somatomotor and Default Mode networks, also exhibit densely clustered, anatomically consistent embeddings. The bilinear low-rank model also produces hub-like clusters that appear aligned with connectivity, but this relative clustering is less aligned with cortical anatomy.

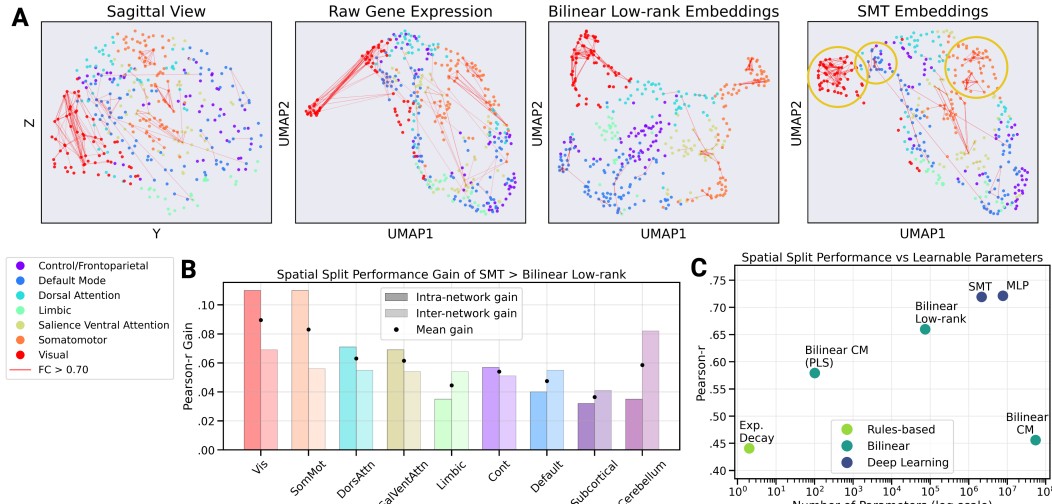

Figure 5: **Model embeddings, non-linear gain, and parameter count analysis. [A]** Anatomical coordinates visualized alongside raw gene expression, Bilinear Low-rank and SMT embeddings using UMAP (subcortex and cerebellum omitted). **[B]** Gain in Pearson-*r* for subnetwork specific test metrics for SMT vs. Bilinear Low-rank model. **[C]** Spatial split test performance plotted as a function of parameter count (log-10 scale) for select linear and non-linear methods.

**Ablation and hyperparameter experiments.** Validation set hyperparameter experiments and post-hoc ablations of the SMT architecture are detailed in Section A.5. These analyses reveal that SMT performance is broadly robust to the choice of tokenization strategy, however, the addition of ALiBi positional slopes consistently improves performance across tokenization variants. We also evaluate eight target augmentation protocols and find that introducing distributional batches of population-level edges during training—particularly those with strong positive or negative edge targets—enhances generalization under both random and spatial splits, with gains of up to Pearson-$r$ 0.05 in strong connectivity ranges (Figure 10). Sorting strategies and alternate target augmentation protocols are core areas of future development, which may support further performance gains of the SMT over MLP despite SMT's reduced parameter count.

**Parameter count analysis.** Figure 5C depicts test set generalization performance under the spatial split protocol as a function of the total number of learnable parameters. Increasing model capacity generally yields a log-linear monotonic improvement in performance, with a plateau observed among non-linear architectures at approximately $10^7$ parameters. Notably, the binning strategy employed by the SMT reduces the parameter count from roughly 7M in the MLP baseline to about 2M with 60 gene bins. At just 80K parameters, the bilinear low-rank model remains competitive, suggesting a potential intermediate regime for compact non-linear transformer-based architectures with preserved performance. In contrast, the bilinear CM model trained on the full gene set contains approximately 55M parameters with massively reduced accuracy, indicating that performance is not solely attributable to parameter count scaling.

## 6 Discussion

A central finding of this work is the significant performance leap achieved by non-linear models (both MLP and SMT) over traditional linear approaches for the genomics-to-connectomics decoding problem. While the absolute performance difference between SMT and MLP is modest, we present them not as competitors but as exemplars of two distinct deep learning strategies for this novel problem. The MLP establishes a powerful, high-performance baseline, confirming the value of non-linear modeling. The SMT, in contrast, serves as a biologically-informed framework whose modular architecture offers unique advantages for scientific inquiry. Its sequential design allows for direct hypothesis testing of biological priors, such as genomic organization via TSS-sorting. Finally, SMT's architecture is naturally suited for future integration with large-scale genomic foundation models [32, 48, 19], offering a clear path for advancing the field.

While this work establishes a new framework, we acknowledge several key limitations. First, the AHBA dataset, despite being the field's gold standard, is derived from only six donors, which may not fully capture population-level transcriptomic diversity. Second, while our non-linear models significantly outperform linear baselines, the modest performance gap between the MLP and SMT highlights that further architectural innovation is needed to fully leverage biological priors. Our qualitative analysis of SMT's embeddings (Fig 5) and attention weights (Fig 13) offers a promising but preliminary step toward interpretability; a systematic validation of these learned gene-set contributions is a critical future direction. Looking ahead, this framework opens several exciting avenues: applying these models to larger and more diverse transcriptomic datasets, integrating individual genetic variations such as polygenic risk scores [49], and extending the models to predict patient-specific connectome alterations [50, 39, 51]. Ultimately, moving from population-level mapping to individualized prediction [52] will be the key to unlocking the clinical potential of this approach for understanding the molecular basis of brain disorders.

**Acknowledgements.** This work was supported by NIH grant R00MN128772 to E.V. and the DOD National Defense Science and Engineering Graduate fellowship. We thank Dhivya Srinivasan, M.S., Theodore D. Satterthwaite, MD, Tien Tong, Ph.D., and Matthew Cieslak, Ph.D., for providing insightful scientific feedback on the study design as well data access and pre-processing assistance. BioRender software was used for figure creation.

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

# A  Technical Appendices and Supplementary Material

## A.1  Dataset Processing

**Allen Human Brain Atlas.**  Transcriptomic data is preprocessed from the Allen Human Brain Atlas (AHBA) using the *abagen* toolbox [53], following a set of best-practice steps to standardize expression across donors and reduce noise from technical and biological sources. Unless otherwise specified, all processing steps follow established workflows [26, 3, 21].

*Within-Donor Normalization.* Gene expression values for all tissue samples are normalized within each donor to account for individual differences in signal scale or distribution. A scaled robust sigmoid transformation is applied to each gene expression vector $x_g$, defined across regions for a given donor:

$$x_{\text{norm}} = \frac{1}{1 + \exp\left(-\frac{x_g - \text{median}(x_g)}{\text{IQR}(x_g)}\right)}$$

This transformation preserves rank while being robust to outliers. A subsequent min-max rescaling step standardizes the values to the unit interval:

$$x_{\text{scaled}} = \frac{x_{\text{norm}} - \min(x_{\text{norm}})}{\max(x_{\text{norm}}) - \min(x_{\text{norm}})}$$

This two-step procedure is distribution-free and avoids assumptions of normality, unlike traditional z-scoring.

*Sample-to-Region Assignment.* Tissue samples are assigned to regions in a reference parcellation using their MNI coordinates, constrained by hemisphere and cortical/subcortical classification. Expression values for samples mapping to the same region are averaged, resulting in a subject-specific region-by-gene matrix of shape $r \times g$.

*Gene Selection and Filtering.* To reduce noise and retain biologically informative genes, we follow a multi-stage filtering process. First, genes expressed below background noise are excluded. Next, we quantify spatial reproducibility via differential stability, defined as:

$$\Delta_S(g) = \frac{1}{\binom{N}{2}} \sum_{i=1}^{N-1} \sum_{j=i+1}^{N} r(B_i(g), B_j(g))$$

where $B_i(g)$ is the regional expression profile of gene $g$ for donor $i$, and $N = 6$ is the number of AHBA donors. Genes with high $\Delta_S(g)$ show consistent regional variation across brains. We use the strictest threshold recommended for the Schaefer-400 parcellation, retaining a final set of 7,380 genes per region.

*Cross-Donor Aggregation.* After normalization and filtering, gene expression values are averaged across donors to produce a population-level matrix. To address incomplete sampling across hemispheres, we employ *abagen*'s mirror-and-interpolate strategy: missing regions are imputed using their contralateral counterparts and smoothed via weighted averaging over the 10 nearest neighbors in the source hemisphere. This yields a complete $r \times g$ matrix aligned to the target parcellation, suitable for downstream modeling.

See [53] for licensing.

**UK Biobank.**  rs-fMRI scans were acquired on a Siemens Skyra 3T scanner (TR = 735 ms, TE = 39 ms, multiband factor = 8, 2.4 mm isotropic resolution, scan duration = 6:10 min) [22]. Preprocessing followed the UKBB minimal pipeline [22] and was further refined using XCP-D [54], implemented in Nipype [55]. This included the removal of non-steady-state volumes, notch filtering of motion regressors, global signal regression, despiking via `3dDespike`, and bandpass filtering (0.01–0.1 Hz) applied to both data and confounds [56, 57]. Population averaging followed the parcellation of individual rs-fMRI scans into connectivity matrices. The S456 atlas used can be found at `https://github.com/PennLINC/AtlasPack/blob/main/tpl-fsLR_atlas-4S456Parcels_dseg.json`.

**Human-Connectome-Project Young Adult (HCP-YA).**  The HCP-YA dataset contains rs-fMRI data processed identically to the UKBB dataset in the S456 parcellation. Despite substantial demographic shift (n=1065; mean age = 29; age range = 22-35; 54% female) between cohorts, we

observe a stable backbone connectivity structure in the population average connectomes between UKBB and HCP (Pearson-$r$ = 0.90, Figure 4). Further details for the HCP-YA cohort can be found at https://pennlinc.github.io/AI2D/docs/datasets/HCP-YA/ [23, 58, 59] .

**Max Planck Institut Leipzig Mind-Brain-Body Dataset (MPI-LEMON).** MPI-LEMON is a comprehensive neuroimaging dataset (n = 136; age range = 20-30; 72% male). The MPI dataset uses an unsupervised voxel-level clustering approach to generate parcellations at various scales including 183, 391, and 729 region resolutions from which functional connectivity can be computed. MPI-LEMON comes with paired, minimally processed AHBA gene expression data at each resolution. Processing was done with the recommended *abagen* [53] parameters. Bi-hemispheric imputation was not done for missing regions in this dataset, which differs from the paired AHBA dataset used for UKBB. See Jimenez-Marin et al. [24] for further details, licensing, and data access.

## A.2 Experimental Details

**Human reference genome ordering.** Genes are sorted by their transcription start site (TSS) in the SMT tokenization procedure. For each protein-coding gene, the earliest TSS is selected across all transcripts, regardless of strand orientation. Genes are then ordered globally—first across chromosomes, then within each chromosome—yielding a biologically grounded input sequence. This ordering is intended to enhance interpretability and biological plausibility for SMT. The reference genome used is available at ncbi.nlm.nih.gov/datasets/genome/GCF_000001405.40. Implementation details are provided in GeneEx2Conn/data/enigma/gene_lists/human_refgenome_README.md

**Parcel centroids.** Spatial coordinates for brain regions are used throughout our analyses. For each parcellation, we determine the centroid of each anatomical region using provided metadata files. Coordinates are defined in the Montreal Neurological Institute (MNI) standard space. For the MPI-LEMON dataset, these centroids are available via the original release [24]. For the UKBB dataset, we provide a reformatted metadata file directly at /data/UKBB/atlas-4S456Parcels_dseg_reformatted.csv. Centroids for this custom parcellation were computed using nilearn.plotting.find_parcellation_cut_coords.

**Stratified metrics ranges.** Brain region centroids serve two main purposes in our analyses: (1) as input features for models incorporating spatial information, such as the SMT with [CLS] token, and (2) to stratify model performance by inter-regional distance. We define short-, mid-, and long-range Pearson correlations by dividing the brain's maximum inter-regional distance (180 mm) into three equal intervals: short-range ($<$ 60 mm), mid-range (60–120 mm), and long-range (120–180 mm). In addition to these distance-stratified metrics, we also compute performance over connection strength categories: strong positive ($r > 0.3$), strong negative ($r < -0.3$), and weak connections ($-0.3 \leq r \leq 0.3$).

## A.3 Model Optimization

**Hyperparameter selection.** To select the optimal hyperparameter configurations for each model in Table 1 and Table 2, we perform nested inner cross-validation using random sampling from a hyperparameter grid defined from wider grid searches and manual fine-tuning. For each model, 3–6 hyperparameter combinations are sampled, with the exact number depending on the model's complexity and the size of the grid. Each model also contains a best default parameters configuration, which is guaranteed to be sampled in addition to the randomly sampled configurations. The combination yielding the lowest validation loss on a dedicated subset of validation nodes, is then used for full training and performance evaluation on the test set. Full hyperparameter grids for all models are available in our repo at /GeneEx2Conn/models/configs. All experiments are run on A100 or H100 NVIDIA GPUs on a high-performance compute cluster, taking no more than 1 hour per fold.

**Optimization details.** All gradient-based models, including both linear baselines and MLP architectures, are trained using the AdamW optimizer. The mean squared error (MSE) between predicted and ground-truth connectivity values are used as the training loss. We perform model selection based on validation-set performance across cross-validation folds. All model architectures and hyperparameter configurations can be found in our repo, github.com/neuroinfolab/GeneEx2Conn, under /models/configs.

**Rules-based baselines.**    We adopt several simple rules-based baselines for predicting inter-regional connectivity $Y_{ij}$ from spatiomolecular properties. The *exponential decay* model assumes decay of connectivity strength with Euclidean distance $d_{ij}$, given by $Y_{ij} = \text{SA}_\infty + (1 - \text{SA}_\infty) \exp(-d_{ij}/\text{SA}_\lambda)$, where $\text{SA}_\infty$ and $\text{SA}_\lambda$ control the asymptotic offset and decay rate [17].

**Bilinear Connectome Model.**    Optimizing gene-gene interaction weights $O$ is challenging due to its quadratic scaling with the number of genes and the large combinatorial space it spans. Kovács et al. [11] restrict their analysis to a subset of 19 innexin genes and estimate $O$ by minimizing $\|\text{vec}(Y) - (X \otimes X)\text{vec}(O)\|_2^2 + \alpha\|\text{vec}(O)\|_2^2$. We adopt a more scalable formulation that minimizes $\min_O \|Y - XOX^\top\|_F^2 + \lambda\|O\|_F^2$, using a closed-form solution described in Section A.4, enabling efficient learning of interaction matrices on the full gene set or on dimensionality-reduced features.

**Bilinear Low-rank model.**    To reduce parameter count and enforce symmetry in the predicted connectome, we further simplify the bilinear model by exploiting the symmetry of $Y$, and optimizing a shared projection matrix $\hat{E} \in \mathbb{R}^{d \times k}$. This yields the objective $\min_{\hat{E}} \|Y - X\hat{E}\hat{E}^\top X^\top\|_F^2 + \lambda\|\hat{E}\|_F^2$, which preserves representational power while significantly lowering computational complexity.

**Partial Least Squares (PLS).**    PLS simultaneously learns projections for inputs $X$ and outputs $Y$ to maximize covariance in a shared latent space, providing a powerful framework for decomposing high-dimensional, collinear data into low-dimensional latent components [60]. The general form of the PLS model can be written as $X = TP^\top + F$ and $Y = UQ^\top + E$, where $P \in \mathbb{R}^{g \times k}$ and $Q \in \mathbb{R}^{r \times k}$ are projection matrices, $T, U \in \mathbb{R}^{r \times k}$ are the corresponding score matrices, and $E, F$ are error terms. Each latent component $k$ of $P$ and $Q$ is learned by maximizing $\max_{p,q}(Xp)^\top(Yq)$, followed by solving the regression $T\beta = U$, yielding the overall model $Y = XP^\top\beta Q^\top$.

**PLS-based encoder-decoder adaptation.**    To adapt PLS for our edge-wise prediction task, we reformulate it into an encoder-decoder framework. We first use its learned projections to create region-level embeddings $t_i = XP^\top$ from $P$ learned on a training set $(X_{\text{train}}, Y_{\text{train}})$. Connection strength is then predicted via a bilinear decoder by minimizing the mean squared error between predicted and observed connectivities, $\min_O \frac{1}{r^2} \sum_{i,j=1}^r (Y_{i,j} - t_i^\top O t_j)^2$. The latent dimensionality $k$ and regularization parameter $\lambda$ are selected via cross-validation.

**Multilayer Perceptron.**    The input to the fully-connected MLP is of shape $2 \times 7380$ with 2-4 hidden layers. Each hidden layer includes batch normalization, dropout, and ReLU activations. The model is trained using the AdamW optimizer with weight decay for regularization. All hyperparameters, including network depth, hidden layer size, dropout rate, and learning rate, are selected via cross-validation to mitigate overfitting given the high-dimensional input and large parameter space.

## A.4 Spatiomolecular Null Shuffle Details

**Implementation.** Null brain maps were developed to address inflated false-positive rates that arise when comparing spatially structured brain data. Instead of naive permutations, spatially informed spin methods generate surrogate maps that preserve spatial autocorrelation while disrupting correspondence with true data, enabling statistical null distributions that isolate the contribution of spatial structure.

We adopt the spin-based procedure of Váša et al. [44], which projects cortical parcel centroids onto a sphere, applies a random rotation, and reassigns each parcel's data to the region it lands on. This method preserves inter-hemispheric contiguity and has been shown to be robust across parcellation resolutions [16]. For subcortical structures, we instead apply the spin (more formally a shuffle) directly in coordinate space for the subcortex and cerebellum, ensuring bilateral symmetry, to overcome the non-spherical topology of the subcortex.

Even with modifications for subcortical structures, a naive spatial spin, though preserving distance-based spatial autocorrelation, may disrupt the genetic autocorrelation structure of the transcriptome. Genetic autocorrelation reflects the principle that regions with similar gene expression profiles tend to exhibit stronger functional connectivity. This pattern partially overlaps with spatial proximity, but may include more complex distance-based molecular relationships.

To quantify transcriptomic autocorrelation, we examine how correlated gene expression (CGE) decays with distance. We bin all region pairs into 5mm intervals based on Euclidean distances and compute the mean CGE within each bin, yielding a CGE–distance decay curve. The canonical curve observed in human data [61] and across species [6] typically follows an exponential decline. Notably, we observe a secondary rise in CGE around 120mm, likely driven by long-range associations between cytoarchitectonically similar regions. Negative CGE values at larger distances often correspond to cortico-subcortical interactions [21].

---

**Algorithm 1** CGE-Matched Spatial Null Brain Map Generation

---

1: **Input:** True transcriptome $X^{\text{true}} \in \mathbb{R}^{r \times g}$, region coordinates $C \in \mathbb{R}^{r \times 3}$, number of spins $N$, number of top spins to return $K$
2: **Define:** Exponential CGE curve $f(d) = \text{SA}_\infty + (1 - \text{SA}_\infty)e^{-d/\text{SA}_\lambda}$
3: **Define:** 3rd-order polynomial CGE curve $g(d) = a_1 d^3 + a_2 d^2 + a_3 d + a_4$
4: Compute true CGE vs. distance parameters $(\text{SA}_\lambda^{\text{true}}, \text{SA}_\infty^{\text{true}}, \{a_1^{\text{true}}, \ldots, a_4^{\text{true}}\})$ from $X^{\text{true}}$ and $C$
5: **for** $i = 1$ to $N$ **do**
6:     **(a) Cortical Spin:** Project cortical coordinates $C_{\text{ctx}}$ to the unit sphere
7:     Generate rotation matrix $R_i \in \text{SO}(3)$ and compute $C_i^{\text{rot}} = C_{\text{ctx}} R_i$
8:     Construct bijective mapping $\pi_i^{\text{ctx}} : \{1, \ldots, r_{\text{ctx}}\} \to \{1, \ldots, r_{\text{ctx}}\}$ based on Váša et al. [44]:
   - Initialize unassigned set $\mathcal{U} = \{1, \ldots, r_{\text{ctx}}\}$
   - While $\mathcal{U} \neq \emptyset$:
      1. Select the most distant pair $(p, q) \in \mathcal{U}$
      2. Assign new index: $\pi_i^{\text{ctx}}(p) = \arg\min_j \|C_i^{\text{rot}}[p] - C_{\text{ctx}}[j]\|_2$, likewise for $q$
      3. Remove $p, q$ from $\mathcal{U}$
9:     **(b) Subcortical Spin:** Define $\pi_i^{\text{sub}}$ by applying a symmetric shuffle on subcortical coordinates
10:     Construct null matrix: $X_i^{\text{null}} = \texttt{concat}(X_{\text{ctx}}[\pi_i^{\text{ctx}}], X_{\text{sub}}[\pi_i^{\text{sub}}])$
11:     Compute CGE vs. distance from $X_i^{\text{null}}$ and $C$
12:     Fit $f_i(d)$ to extract $(\text{SA}_\lambda^i, \text{SA}_\infty^i)$
13:     Fit $g_i(d)$ to obtain $(a_1^i, a_2^i, a_3^i, a_4^i)$
14:     Compute total reassignment cost $c_i = \sum_{j=1}^{r_{\text{ctx}}} \|C_{\text{ctx}}[j] - C_{\text{ctx}}[\pi_i^{\text{ctx}}(j)]\|_2$
15: **end for**
16: Standardize each CGE parameter and cost across null spins using z-scores
17: Compute total standardized error for each spin $i$:

$$e_i = \left| z(\text{SA}_\lambda^i) - z(\text{SA}_\lambda^{\text{true}}) \right| + \left| z(\text{SA}_\infty^i) - z(\text{SA}_\infty^{\text{true}}) \right| + \sum_{j=1}^{4} \left| z(a_j^i) - z(a_j^{\text{true}}) \right| + \left| z(c_i) - z(c^{\text{true}}) \right|$$

18: Rank spins by ascending $e_i$
19: **Return:** Top $K$ lowest-error null transcriptomes $\{X_{i(1)}^{\text{null}}, \ldots, X_{i(K)}^{\text{null}}\}$

---

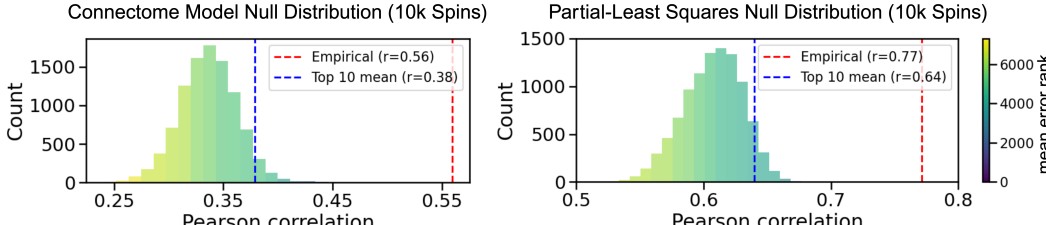

Figure 6: **Null distributions of Connectome Model and Partial Least Squares.** Each histogram shows the distribution of Pearson correlation between predicted and observed connectomes across 10,000 spatially spun/shuffled gene expression matrices. Left: Connectome Model (CM). Right: Partial Least Squares (PLS). Bars are colored by the average mean error rank across 25 bins (darker = better rank). Red dashed lines mark empirical performance on the true gene expression matrix; blue dashed lines indicate the mean performance using the 10 lowest error nulls.

Figure 3 illustrates that spin-based nulls generated by Algorithm 1 produce highly variable CGE–distance curves, ranging from biologically realistic exponential decays to distorted, spatially disordered profiles. To address this, we implement a rejection sampling procedure that selects null transcriptomes best preserving both spatial and transcriptomic autocorrelation, enabling stringent null testing without requiring full null distributions. The left panel of Figure 3 shows the empirical CGE decay from the true brain, while representative outcomes of our procedure are labeled as *"Low-error null spin"* and *"High-error null spin"*.

In our experiments, we set $N = 10{,}000$ spatial spins and select the top $K = 10$ based on Algorithm 1. The full implementation is provided in our repository at `/sim/null.py` within the `generate_null_spins()` function. The procedure for fitting the exponential decay and polynomial parameters are also outlined in this Python file.

**Effect of mean error rank.** Despite the mismatch between features and targets introduced by spatial spinning, we hypothesize that gene expression inputs preserving realistic spatial and genetic structure will yield more realistic connectome predictions. If autocorrelation is indeed predictive of connectivity, null brain transcriptome datasets with more realistic CGE profiles should result in predictions closer to the true value.

We test this by fitting two linear models—the Connectome Model (CM) and Partial Least Squares (PLS)—using 10,000 spatiomolecular null spun/shuffled brains. For each spin, we fit the models using the null gene expression matrix $X'$ and evaluate the Pearson correlation between predicted $\hat{Y}$ and observed $Y$. For PLS, we use 10 latent components (selected via grid search over 0–25 dimensions selecting the elbow point based on Pearson $r$); for CM, we use PCA-reduced gene expression with 27 components (capturing 95% of the variance), enabling efficient training via the closed-form solution outlined below. We train and test on the full dataset as is standard for generating null distributions under null spin tests [16].

Figure 6 indeed confirms that model performance across spins is sensitive to the quality of the CGE profile. Histograms of prediction accuracy (Pearson-$r$) reveal that the top 10 most CGE-consistent nulls produce better fits than the bulk of nulls. This effect is consistent across both models, and supports the idea that realistic gene-distance relationships are a critical determinant of null predictive accuracy.

**Closed form solution for the Connectome Model.** We solve the ridge-regularized bilinear regression problem:

$$\min_{O} \; \|Y - XOX^\top\|_F^2 + \lambda \|O\|_F^2,$$

where $X \in \mathbb{R}^{n \times d}$, $Y \in \mathbb{R}^{n \times n}$, $O \in \mathbb{R}^{d \times d}$, and $\lambda > 0$. $\lambda$ is set following Kovács et al. [11]. Using $\|A\|_F^2 = \mathrm{Tr}(A^\top A)$, we expand the objective:

$$\mathcal{L}(O) = \mathrm{Tr}(Y^\top Y) - 2\,\mathrm{Tr}(OX^\top YX) + \mathrm{Tr}(O^\top X^\top XOX^\top X) + \lambda\,\mathrm{Tr}(O^\top O).$$

Letting $A = X^\top X$ and $M = X^\top Y X$, we simplify:
$$\mathcal{L}(O) = \text{const} - 2\,\text{Tr}(OM) + \text{Tr}(O^\top AOA) + \lambda\,\text{Tr}(O^\top O).$$

Taking the gradient with respect to $O$ gives:
$$\nabla_O \mathcal{L} = -2M + 2AOA + 2\lambda O.$$

Setting the gradient to zero and rearranging:
$$AOA + \lambda O = M.$$

Assuming $A$ is symmetric and positive definite (e.g., after PCA), we pre- and post-multiply by $(A + \lambda I)^{-1}$ to obtain:
$$O^* = (A + \lambda I)^{-1} M (A + \lambda I)^{-1},$$
or equivalently,
$$O^* = (X^\top X + \lambda I)^{-1} X^\top Y X (X^\top X + \lambda I)^{-1}.$$

This formulation enables vast speedups over Kovács et al. [11] original Kronecker formulation, enabling fitting of the Connectome Model for many iterations and for larger dimension, $d$.

**Train-test split with spatiomolecular null comparison.** As described in 4, each model in Table 1 is tested across 10 random or spatial four-fold splits. For each split, a corresponding unique spatiomolecular null gene expression matrix (from the top 10 out of 10,000) is used to refit the model and predict held out test set. This results in 40 test performance metrics for the true gene expression and 40 test performance metrics for the null gene expression. Aggregate metrics are computed to compare the true vs null model performance. This procedure circumvents the need to refit the model thousands of times using many, potentially suboptimal, null spins.

**MLP/SMT performance with coordinates & null gene expression.** In Table 1 we observe a substantial true vs. null gap in Pearson-$r$, however, this gap is markedly smaller for the SMT w/ [CLS] under the random split. We posit that this trend is due to the fact that the MLP w/ coords and the SMT w/ [CLS] have access to the true coordinates in both the true and null case. Thus, Euclidean coordinates alone might be predictive of connectivity even without molecular information. To deduce the effect of spatial position alone, we fit several variations of null models in Table 3.

Table 3: Null model performance comparison with varying feature access on UKBB dataset (mean and standard deviation over 10 random 4-fold splits)

| Model | Features | Pearson-r | Short r | Mid r | Long r | R² | MSE | Geodesic |
|---|---|---|---|---|---|---|---|---|
| MLP | coords | .72 ± .06 | .76 ± .06 | .67 ± .07 | .63 ± .10 | .52 ± .09 | .016 ± .003 | 13.12 ± 1.01 |
| MLP | permuted + coords | .29 ± .07 | .28 ± .08 | .20 ± .06 | .16 ± .09 | -.05 ± .11 | .035 ± .004 | 16.45 ± .80 |
| MLP | SM + coords | .47 ± .08 | .48 ± .08 | .39 ± .08 | .36 ± .11 | .11 ± .13 | .030 ± .005 | 14.31 ± 1.2 |
| SMT w/ CLS | permuted + coords | .68 ± .06 | .70 ± .07 | .64 ± .06 | .61 ± .10 | .41 ± .11 | .020 ± .004 | 11.79 ± .92 |
| SMT w/ CLS | SM + coords | .71 ± .05 | .73 ± .05 | .67 ± .05 | .65 ± .08 | .47 ± .09 | .018 ± .003 | 11.52 ± .84 |

SM indicates gene expression has been permuted under the spatiomolecular null procedure. Permuted indicates the gene expression is a purely random permutation reassignment.

The MLP trained with coordinates alone demonstrates that spatial position is a strong predictor of functional connectivity, achieving Pearson-$r > 0.7$. This likely accounts for the similarly high performance of the SMT w/ [CLS] model under the random split spatiomolecular null. Aside from architectural differences, the only distinction between these models is that the SMT receives additional (but spatially spun) gene expression input. Still, the SMT w/ [CLS] matches the performance of MLP w/ coords across all metrics. The difference observed between the permuted gene expression and spun gene expression highlights the stringency of the spatiomolecular null as argued in A.4. Overall, an MLP trained just on coordinates is effective at connectome reconstruction.

These results reinforce that any model exceeding the SMT with [CLS] null reflects learning of true transcriptomic patterns, rather than spatially autocorrelated structure. Notably, the SMT w/ [CLS] architecture appears to leverage spatial information more explicitly than the MLP, which has reduced ability to isolate positional cues from the high-dimensional input. Attention heads in Figure 13 support strong emphasis of the [CLS] token during learning. In the UKBB dataset, several learned models surpass this stringent null baseline (see Table 1), underscoring their capacity to extract meaningful molecular information beyond what is encoded by Euclidean distance.

## A.5 Additional Experiments

**Hyperparameter experiments.** To isolate the effect of key design choices in the SMT architecture, we conduct a series of hyperparameter experiments using 10 fixed inner cross-validation splits with the UKBB dataset under our random train-test split. These experiments alter each hyperparameter independently while fixing the default SMT configuration. Figure 7 summarizes the effects of four components: token encoder dimension, token encoder output dimension (number of genes per token), ALiBI slopes, and target augmentation probability. Table 4 displays the hyperparameter grid for the Spatiomolecular Transformer with default parameters bolded.

*Inner-CV Experiments*

Token encoder dimension performs best at 60, grouping larger gene chunks into fewer tokens. This improves speed (reducing quadratic attention cost) with minimal performance trade-off. Advances in efficient attention may eventually allow single-gene tokenization at scale.

Fixing the token encoder dimension to 60 dimension gene chunk bins and varying the output dimensionality after the transformer shows relative stability up to an output dimensionality of 3 per token. A token encoder output dimensionality of 10 is optimal on the validation set.

Table 4: Hyperparameter search space for the Spatiomolecular Transformer. Default values in **bold**.

| Hyperparameter | Values |
| --- | --- |
| token_encoder_dim | [20, **60**, 180] |
| d_model | [64, **128**] |
| encoder_output_dim | [5, **10**] |
| use_alibi | [**True**, False] |
| nhead | [2, **4**] |
| num_layers | [2, **4**] |
| deep_hidden_dims | [[256,128], **[512,256,128]**] |
| transformer_dropout | [0.1, **0.2**] |
| dropout_rate | [0.1, **0.2**, 0.3] |
| learning_rate | [0.00009, **0.0001**] |
| weight_decay | [**0.0001**, 0.001] |
| batch_size | [**512**, 1024] |
| aug_prob | [0, 0.1, **0.3**] |
| aug_style | [linear_decay, linear_peak, curriculum_swap_constant, **curriculum_swap_linear_decay**] |
| epochs | [90, **110**] |
| num_workers | [**2**] |
| prefetch_factor | [**4**] |

ALiBi slopes show a simple and clear trend improving both Pearson-*r* and MSE when incorporated into the MHSA mechanism, pointing towards the utility of multi-head representations of the input sequence. In the next section we explore the effect of alternate sorting strategies under the random and spatial split. Modest amounts of target augmentation probability 0.1-0.3 are optimal over more aggressive strategies. This is tested for the standard linear decay setting. We explore target-side augmentation strategies further in the subsequent sections.

In general it is worth noting that differences between hyperparameters are nominal indicating nuanced effects of each on overall learning, as well as potential combinatorial effects.

*Tokenization strategy*

To evaluate the effect of our TSS based tokenization strategy we conduct a post-hoc experiment, where all hyperparameters are fixed to default. We then systematically vary the gene bin ordering and the size of the bins. For each gene bin size, $k$, genes are either binned together randomly, by TSS, or by mean global expression patterns. The mean expression based setup may facilitate the co-embedding of genes based on coexpression and is commonly used in single-cell transformer based methods [32]. Figure 8 shows changes in performance when varying the token size and tokenization strategy. The first notable effect

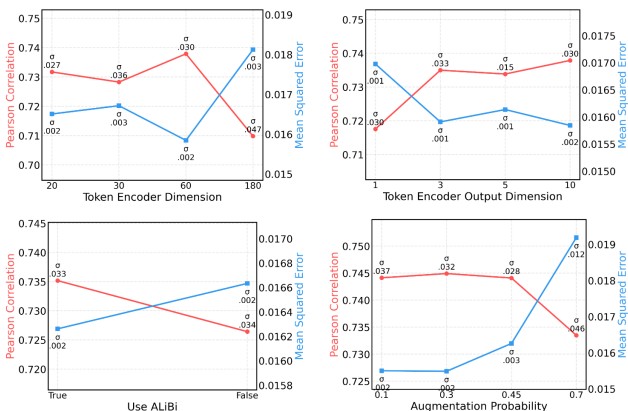

Figure 7: **Validation set hyperparameter experiments for SMT.** Performance varies with token dimensionalities, ALiBi slopes, and augmentation probabilities.

is that the SMT performs best on both splits with gene groups of 60. There is no clear added benefit to smaller gene bins, which aligns with the inner-CV experiments in Figure 7. Even when genes are sorted and binned completely randomly, with no biological prior information, SMT performance does not drop. Performance only drops when ALiBi slopes is removed from the model. This points towards a robustness of the *value projection* transformer style model and that there may be alternate architectural modifications that would introduce more useful biological priors into the SMT. ALiBi

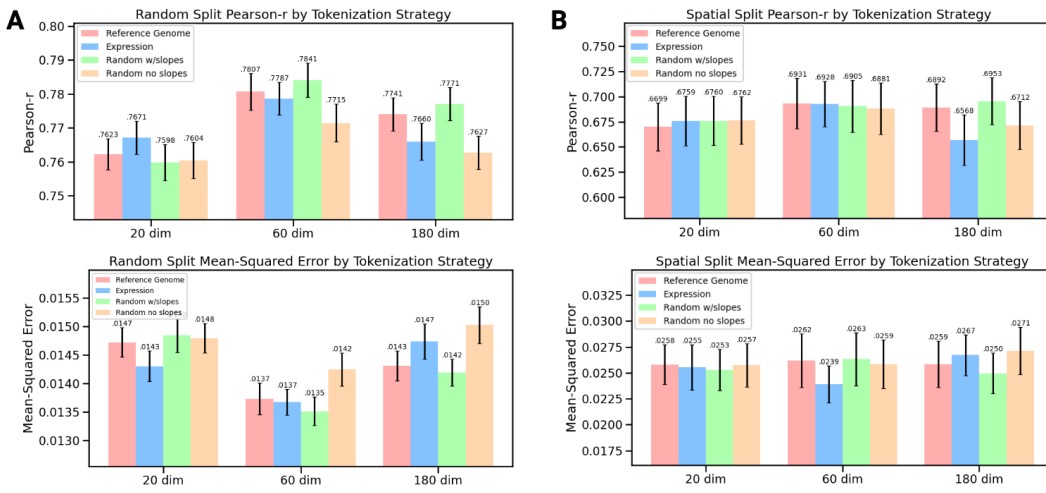

Figure 8: **Test set tokenization strategy analysis.** TSS-based, mean global expression-based, and randomly sorted tokenization with and without ALiBi slopes are evaluated on the test set.

slopes, however, seems to have a positive effect on model performance likely due to its encouraged hierarchical representation of the gene expression sequence agnostic of tokenization sorting strategy.

*Target-side augmentation strategy*

Given the data-scarce domain of population-level transcriptome-to-connectome prediction, we introduce a distributional target-side augmentation strategy to improve generalization. Our approach combines ideas from curriculum learning [42, 25] and synthetic minority oversampling techniques [43], modifying training batches throughout the learning process by injecting signal from the full population distribution.

Algorithm 2 outlines this procedure, which defines three possible augmentation paths: (i) using targets from the population-average connectome $Y$ (i.e., no augmentation), (ii) randomly replacing targets in the current batch with individual-level targets from the population, and (iii) replacing targets only if their original population value satisfies $|y| > \theta$, where we fix $\theta = 0.3$. This threshold ensures that only strong positive or negative connections are overrepresented during training—corresponding to a minority class in typical connectome distributions (see the density function in the bottom right of Fig. 1).

---

**Algorithm 2** Distributional Target-Side Augmentation Protocol

---

**Require:** Current mini-batch $\mathcal{B} = \{(X_i, y_i)\}_{i=1}^{B}$, epoch $e$, total epochs $E$, augmentation style `aug_style`, probability schedule $p(e)$, population matrix $\mathcal{Y}_{\text{pop}}$
1: Draw $r \sim \mathcal{U}(0, 1)$
2: **if** $r < p(e)$ **then**
3:     **if** `aug_style = curriculum_swap` **then**
4:         Identify indices in $\mathcal{D}_{\text{train}}$ where $|y| > \theta$
5:         Sample new batch $\mathcal{B}$ from strong edges
6:         Replace all $y_i$ in $\mathcal{B}$ with subject-level values from $\mathcal{Y}_{\text{pop}}$
7:     **else**
8:         **for** each target $y_i$ in $\mathcal{B}$ **do**
9:             Replace $y_i$ with a subject-level value from $\mathcal{Y}_{\text{pop}}$
10:         **end for**
11:     **end if**
12: **else**
13:     No augmentation is applied; original batch is used
14: **end if**
15: Compute loss: $\mathcal{L}_{\text{MSE}} = \frac{1}{B} \sum_{i=1}^{B} \|f_\theta(X_i) - y_i\|^2$

---

The augmentation schedule $p(e)$ determines the probability of performing target-side augmentation at epoch $e$. Various scheduling strategies—such as linear increase, decrease, constant, and peak—are visualized in Fig. 9. This mechanism allows us to inject task-relevant population signal at different stages of training. In our main experiments, we select the augmentation strategy using cross-validation, and Fig. 10 analyzes performance differences across augmentation protocols. For all styles, the maximum augmentation probability is fixed at $p = 0.3$, based on inner cross-validation results using linear decay.

Nearly all augmentation strategies improve model performance with respect to Pearson correlation

Figure 9: **Augmentation schedules defining** $p(e)$ **over training epochs.**

and mean squared error (MSE). Gains are especially pronounced when evaluating Pearson-$r$ restricted to strong positive edges ($y > 0.3$), where both constant and linearly decaying augmentation schedules yield an improvement of $\sim 0.05$. This effect is intuitive, as minority class oversampling emphasizes edges with strong signal, which are otherwise underrepresented. Although global metric gains are moderate, consistent gains suggest that curriculum-guided target-side augmentation improves model robustness across both random and spatial data splits, and should be considered as a general-purpose regularization strategy in transcriptome-connectome modeling tasks in future work.

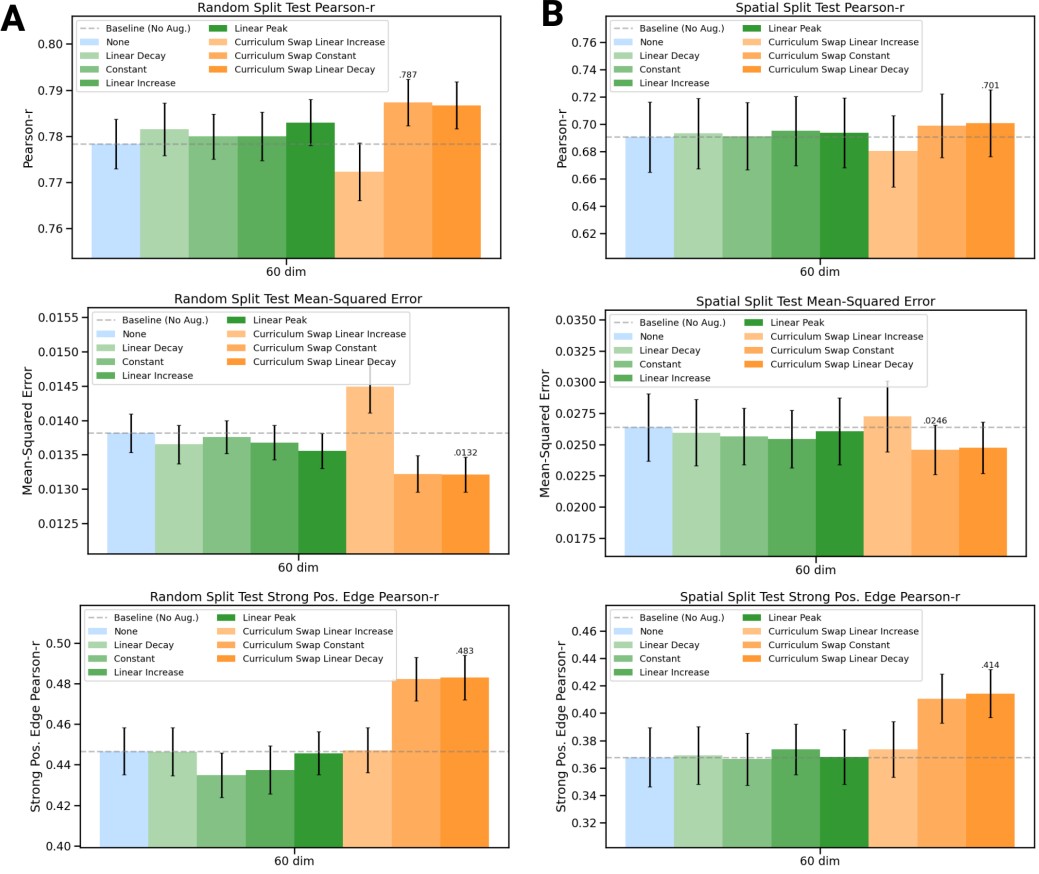

Figure 10: **Distributional target-side augmentation analysis.** Augmentation style test-set performance for Pearson-$r$, MSE, and strong-positive Pearson-$r$ (>0.3) for **[A]** random split and **[B]** spatial split.

### A.6 Replication Analysis

**Intra-resolution replication analysis.** For all resolutions of the MPI-LEMON and HCP dataset, the SMT exceeds its corresponding null estimate (Table 5 and 7) under both the random and spatial split setting. The raw performance values are lower than those reported for UKBB, which could be attributed to a variety of fundamental differences between the datasets such as data quality, processing steps, parcellation style, or weaker signal between the AHBA population and the HCP and MPI-LEMON populations. Alternatively, despite using cross-validated hyperparameter searching strategy, a more optimal search space may exist for the MPI-LEMON dataset as compared to UKBB. Another primary difference between the datasets is that the MPI-LEMON dataset contains more positively skewed values whereas UKBB contains strong signal in both the positive and negative direction.

Tables 6 and Table 8 similarly illustrate performance above the spatiomolecular null but at a smaller margin for MPI-LEMON. Strong spatial effects are present at all resolutions in the null setting. This gap is in line with our SMT w/ [CLS] results from 1 which shows a much smaller gap when true coordinates are introduced to the null model. In sum, Tables 5 through 8 confirm a robust relationship between gene expression and functional connectivity in the HCP-YA and MPI-LEMON dataset, but there may be more weakly aligned genetic signal in the MPI-LEMON dataset or additional noise introduced by technical artifacts.

**Cross-dataset and cross-resolution analysis.** We use a full dataset framework to test cross-dataset and cross-resolution generalization simultaneously. Here, we train an SMT w/ [CLS] model on a source dataset-resolution pair (e.g., UKBB S456) and reconstruct the full connectome of a different dataset-resolution pair (e.g., MPI-LEMON 729). To evaluate whether performance exceeds chance, we benchmark against an SMT model trained on one of the top 10 spatiomolecular spins for the source dataset, keeping spatial coordinates fixed. This model represents the baseline explained by spatial proximity and transcriptomic autocorrelation.

For example, we first train the SMT w/ [CLS] model on the true data and a top $K$ null spin of the UKBB S456. If a model trained on MPI-LEMON 729 reconstructs the UKBB S456 connectome more accurately than the UKBB-based null model, it implies generalization beyond spatial and genetic autocorrelation. Default model hyperparameters are selected prior to application on the target dataset.

Table 5: Multi-resolution test set performance for SMT with random CV on MPI-LEMON and HCP-YA dataset (mean and standard deviation over 10 4-fold splits). (*null*) indicates model trained on 10 lowest-error spatiomolecular null brain maps at each resolution.

| Model | Resolution | Pearson-$r$ | Short $r$ | Mid $r$ | Long $r$ | $R^2$ | MSE | Geodesic |
|---|---|---|---|---|---|---|---|---|
| SMT (*null*) | 183 | $0.26 \pm 0.10$ | $0.16 \pm 0.12$ | $0.17 \pm 0.10$ | $0.24 \pm 0.13$ | $-0.11 \pm 0.23$ | $0.030 \pm 0.005$ | $6.62 \pm 0.87$ |
| SMT | 183 | $0.50 \pm 0.09$ | $0.43 \pm 0.11$ | $0.41 \pm 0.11$ | $0.45 \pm 0.09$ | $0.17 \pm 0.16$ | $0.022 \pm 0.005$ | $5.62 \pm 0.98$ |
| SMT (*null*) | 391 | $0.28 \pm 0.06$ | $0.22 \pm 0.06$ | $0.20 \pm 0.05$ | $0.20 \pm 0.10$ | $-0.06 \pm 0.07$ | $0.025 \pm 0.002$ | $9.19 \pm 0.63$ |
| SMT | 391 | $0.57 \pm 0.05$ | $0.56 \pm 0.06$ | $0.49 \pm 0.06$ | $0.48 \pm 0.08$ | $0.28 \pm 0.08$ | $0.017 \pm 0.002$ | $8.17 \pm 0.92$ |
| SMT (*null*) | 729 | $0.30 \pm 0.05$ | $0.24 \pm 0.05$ | $0.21 \pm 0.05$ | $0.25 \pm 0.07$ | $-0.08 \pm 0.08$ | $0.022 \pm 0.002$ | $11.90 \pm 0.77$ |
| SMT | 729 | $0.54 \pm 0.04$ | $0.53 \pm 0.04$ | $0.45 \pm 0.04$ | $0.45 \pm 0.06$ | $0.23 \pm 0.06$ | $0.016 \pm 0.001$ | $10.67 \pm 0.77$ |
| SMT (*null*) | HCP | $0.27 \pm 0.07$ | $0.28 \pm 0.08$ | $0.21 \pm 0.07$ | $0.20 \pm 0.10$ | $-0.22 \pm 0.11$ | $0.030 \pm 0.004$ | $14.41 \pm 0.87$ |
| SMT | HCP | $0.71 \pm 0.04$ | $0.75 \pm 0.04$ | $0.68 \pm 0.04$ | $0.58 \pm 0.08$ | $0.47 \pm 0.08$ | $0.013 \pm 0.002$ | $10.89 \pm 0.77$ |

Table 6: Multi-resolution test set performance for SMT w/ [CLS] on random split (mean and standard deviation over 10 4-fold splits).

| Model | Resolution | Pearson-$r$ | Short $r$ | Mid $r$ | Long $r$ | $R^2$ | MSE | Geodesic |
|---|---|---|---|---|---|---|---|---|
| SMT w/ CLS (*null*) | 183 | $0.50 \pm 0.09$ | $0.44 \pm 0.12$ | $0.39 \pm 0.11$ | $0.45 \pm 0.12$ | $0.04 \pm 0.18$ | $0.026 \pm 0.004$ | $6.18 \pm 0.93$ |
| SMT w/ CLS | 183 | $0.57 \pm 0.09$ | $0.50 \pm 0.14$ | $0.48 \pm 0.12$ | $0.50 \pm 0.12$ | $0.14 \pm 0.24$ | $0.023 \pm 0.007$ | $5.80 \pm 1.11$ |
| SMT w/ CLS (*null*) | 391 | $0.61 \pm 0.06$ | $0.60 \pm 0.06$ | $0.54 \pm 0.06$ | $0.54 \pm 0.09$ | $0.30 \pm 0.10$ | $0.016 \pm 0.002$ | $7.76 \pm 0.80$ |
| SMT w/ CLS | 391 | $0.68 \pm 0.04$ | $0.68 \pm 0.04$ | $0.61 \pm 0.05$ | $0.61 \pm 0.06$ | $0.42 \pm 0.07$ | $0.013 \pm 0.001$ | $7.61 \pm 0.81$ |
| SMT w/ CLS (*null*) | 729 | $0.66 \pm 0.04$ | $0.65 \pm 0.04$ | $0.59 \pm 0.05$ | $0.59 \pm 0.06$ | $0.38 \pm 0.07$ | $0.013 \pm 0.002$ | $9.63 \pm 0.71$ |
| SMT w/ CLS | 729 | $0.70 \pm 0.04$ | $0.70 \pm 0.03$ | $0.63 \pm 0.03$ | $0.64 \pm 0.06$ | $0.45 \pm 0.07$ | $0.011 \pm 0.001$ | $9.67 \pm 1.03$ |
| SMT w/ CLS (*null*) | HCP | $0.59 \pm 0.05$ | $0.63 \pm 0.05$ | $0.53 \pm 0.06$ | $0.53 \pm 0.10$ | $0.26 \pm 0.09$ | $0.018 \pm 0.003$ | $12.59 \pm 0.72$ |
| SMT w/ CLS | HCP | $0.74 \pm 0.05$ | $0.77 \pm 0.04$ | $0.70 \pm 0.05$ | $0.66 \pm 0.10$ | $0.52 \pm 0.08$ | $0.012 \pm 0.002$ | $11.27 \pm 0.81$ |

Table 7: Multi-resolution test set performance for SMT on spatial split (mean and standard deviation over 10 4-fold splits).

| Model | Resolution | Pearson-$r$ | Short $r$ | Mid $r$ | Long $r$ | $R^2$ | MSE | Geodesic |
|---|---|---|---|---|---|---|---|---|
| SMT (*null*) | 183 | $0.13 \pm 0.10$ | $0.09 \pm 0.08$ | $0.08 \pm 0.10$ | $0.11 \pm 0.14$ | $-0.25 \pm 0.41$ | $0.041 \pm 0.013$ | $7.02 \pm 0.84$ |
| SMT | 183 | $0.38 \pm 0.15$ | $0.35 \pm 0.16$ | $0.31 \pm 0.12$ | $0.27 \pm 0.18$ | $0.02 \pm 0.28$ | $0.032 \pm 0.008$ | $6.27 \pm 0.81$ |
| SMT (*null*) | 391 | $0.19 \pm 0.08$ | $0.15 \pm 0.07$ | $0.11 \pm 0.09$ | $0.08 \pm 0.16$ | $-0.15 \pm 0.11$ | $0.034 \pm 0.006$ | $9.34 \pm 0.94$ |
| SMT | 391 | $0.49 \pm 0.10$ | $0.47 \pm 0.11$ | $0.36 \pm 0.09$ | $0.28 \pm 0.21$ | $0.18 \pm 0.17$ | $0.024 \pm 0.006$ | $8.09 \pm 0.84$ |
| SMT (*null*) | 729 | $0.22 \pm 0.09$ | $0.18 \pm 0.07$ | $0.09 \pm 0.07$ | $0.10 \pm 0.09$ | $-0.16 \pm 0.12$ | $0.031 \pm 0.005$ | $11.98 \pm 0.78$ |
| SMT | 729 | $0.48 \pm 0.09$ | $0.45 \pm 0.06$ | $0.34 \pm 0.09$ | $0.28 \pm 0.14$ | $0.17 \pm 0.11$ | $0.022 \pm 0.004$ | $10.73 \pm 0.83$ |
| SMT (*null*) | HCP | $0.25 \pm 0.09$ | $0.25 \pm 0.10$ | $0.15 \pm 0.08$ | $0.09 \pm 0.10$ | $-0.20 \pm 0.18$ | $0.038 \pm 0.011$ | $13.91 \pm 1.18$ |
| SMT | HCP | $0.63 \pm 0.15$ | $0.63 \pm 0.17$ | $0.55 \pm 0.14$ | $0.44 \pm 0.22$ | $0.20 \pm 0.81$ | $0.024 \pm 0.014$ | $11.13 \pm 1.80$ |

Table 8: Multi-resolution test set performance for SMT w/ [CLS] on spatial split (mean and standard deviation over 10 4-fold splits).

| Model | Resolution | Pearson-$r$ | Short $r$ | Mid $r$ | Long $r$ | $R^2$ | MSE | Geodesic |
|---|---|---|---|---|---|---|---|---|
| SMT w/ CLS (*null*) | 183 | $0.29 \pm 0.13$ | $0.24 \pm 0.13$ | $0.14 \pm 0.14$ | $0.22 \pm 0.13$ | $-0.35 \pm 0.59$ | $0.044 \pm 0.019$ | $7.28 \pm 2.47$ |
| SMT w/ CLS | 183 | $0.32 \pm 0.12$ | $0.27 \pm 0.14$ | $0.19 \pm 0.14$ | $0.34 \pm 0.16$ | $-0.28 \pm 0.39$ | $0.042 \pm 0.015$ | $7.01 \pm 1.43$ |
| SMT w/ CLS (*null*) | 391 | $0.43 \pm 0.10$ | $0.40 \pm 0.10$ | $0.28 \pm 0.13$ | $0.35 \pm 0.15$ | $0.00 \pm 0.19$ | $0.030 \pm 0.006$ | $8.02 \pm 0.81$ |
| SMT w/ CLS | 391 | $0.50 \pm 0.09$ | $0.48 \pm 0.10$ | $0.34 \pm 0.10$ | $0.35 \pm 0.17$ | $0.15 \pm 0.15$ | $0.024 \pm 0.006$ | $7.95 \pm 0.83$ |
| SMT w/ CLS (*null*) | 729 | $0.44 \pm 0.11$ | $0.41 \pm 0.10$ | $0.25 \pm 0.12$ | $0.23 \pm 0.16$ | $-0.08 \pm 0.27$ | $0.028 \pm 0.007$ | $10.19 \pm 0.72$ |
| SMT w/ CLS | 729 | $0.52 \pm 0.09$ | $0.49 \pm 0.07$ | $0.36 \pm 0.12$ | $0.31 \pm 0.22$ | $0.08 \pm 0.21$ | $0.025 \pm 0.007$ | $9.87 \pm 0.88$ |
| SMT w/ CLS (*null*) | HCP | $0.37 \pm 0.11$ | $0.37 \pm 0.12$ | $0.21 \pm 0.12$ | $0.18 \pm 0.13$ | $-0.27 \pm 0.58$ | $0.040 \pm 0.017$ | $13.14 \pm 1.56$ |
| SMT w/ CLS | HCP | $0.57 \pm 0.13$ | $0.57 \pm 0.14$ | $0.45 \pm 0.14$ | $0.47 \pm 0.23$ | $0.17 \pm 0.30$ | $0.026 \pm 0.012$ | $12.05 \pm 2.21$ |

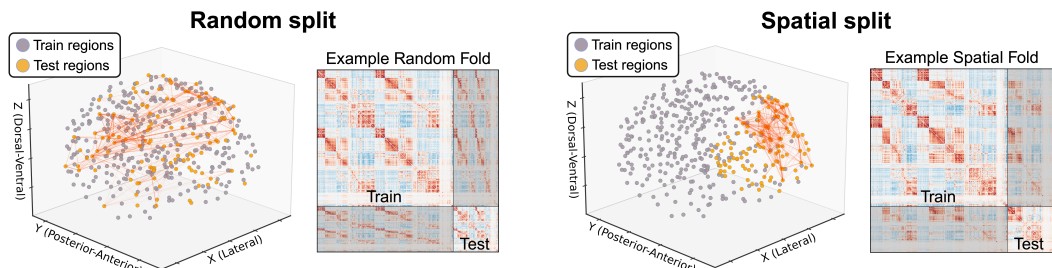

Figure 11: **Example connectome train-test folds for random and spatial splits.** Weighted target edges are visualized in red between test nodes thresholded at $r \geq 0.3$ to sparsify the number of edges.

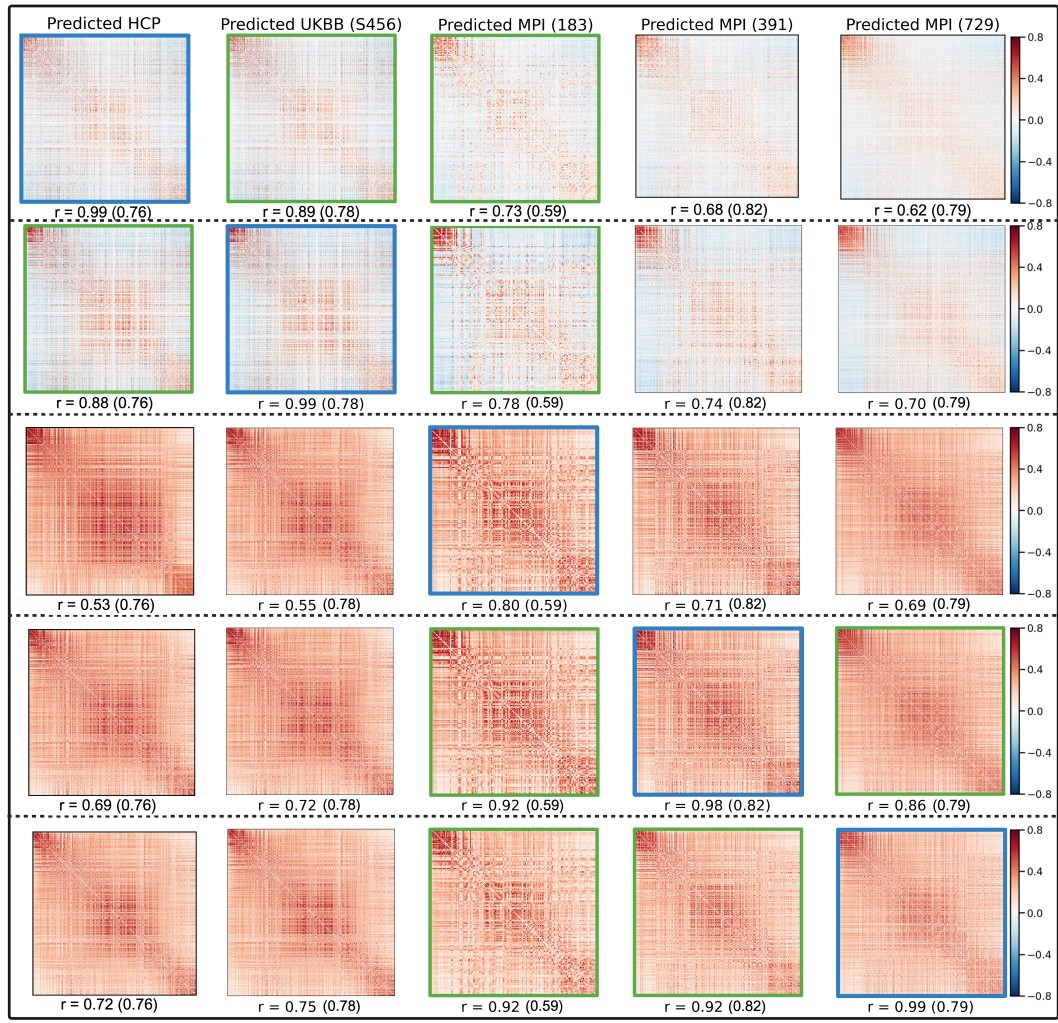

Figure 12: **Cross-dataset generalization of SMT with CLS.** The diagonal shows population-average functional connectomes from UK Biobank (S456), Human Connectome Project (S456), and MPI-LEMON (183, 391, 729 resolutions). Connectomes are sorted by the anterior-posterior axis across datasets and resolutions. Each row shows reconstructions from SMT w/ [CLS] models trained on a source dataset (highlighted with a blue border) and tested on external datasets or resolutions. Pearson correlation between predicted and true connectomes is shown below each matrix, with the source dataset's spatiomolecular null performance estimate in parentheses. Null performance is based on SMT w/ [CLS] models fit on a low-error spatiomolecular null gene expression matrix with true coordinates. Connectomes highlighted with a green border indicate generalization above native SMT [CLS] null chance. (Non-parenthesized Pearson correlations along the diagonal correspond to training fit and thus do not indicate significance over the null. See intra-resolution effects in Tables 5–8).

As shown in Figure 12, across all experiments we observe consistently high reconstruction performance and a backbone connectome structure at all resolutions. Meaningful generalization above the null is observed in a few cases. Both the UKBB and HCP trained models are able to predict MPI-LEMON 183 beyond the spatiomolecular null baseline, suggesting a possible conserved molecular–connectomic relationship across adult populations. Notably, UKBB participants are older on average than those in MPI-LEMON (see Section A.1), supporting the hypothesis that shared transcriptomic gradients contribute to functional brain organization across the adult lifespan. This is further supported by the fact that the HCP trained model significantly predicts MPI-LEMON 183, at a value slightly worse than UKBB, despite more closely aligned demographics between HCP and

MPI-LEMON. HCP to UKBB generalization exceeds null chance, which is expected given the targets of UKBB are highly similar.

We observe that while UKBB and HCP connectomes contain both positive and negative values, MPI-LEMON matrices are strictly positive. This discrepancy may limit the model's ability to capture absolute connectivity values across datasets (as reflected in MSE), though relative connection strengths remain well preserved (as reflected in Pearson-$r$). Parcellation differences likely also play a role: the UKBB parcellation is functionally defined based on resting-state networks, whereas the MPI-LEMON atlas is derived from unsupervised voxel-level clustering.

Within the MPI-LEMON dataset, we find strong generalization across parcellation resolutions. Five cross-resolution reconstructions exceed the null baseline, including one case where a model trained at lower resolution (391) generalizes successfully to a higher resolution (729). These results suggest that while spatial position explains much of the variance, gene expression captures complementary structure. As resolution increases and more training data becomes available, models are better able to learn complex molecular–connectomic relationships.

Our cross-dataset generalization analysis demonstrates that predictive accuracy on UKBB is not an artifact of the particular dataset's properties. There are common spatiomolecular relationships that the SMT effectively learns despite differences in datasets, demographics, imaging acquisition parameters, and parcellation resolutions.

## A.7 Attention Weights Analysis

To interpret how the SMT attends to gene tokens, we extract and average attention weights from the final layer of the trained transformer encoder. For a given region, the input is a sequence of tokenized gene expression values $X \in \mathbb{R}^{\ell \times d}$, where $\ell$ is the number of tokens and $d$ is the embedding dimension. In each layer, $X$ is linearly projected to queries, keys, and values: $Q = XW_Q$, $K = XW_K$, $V = XW_V$, where $W_Q, W_K, W_V \in \mathbb{R}^{d \times d_h}$, and $d_h = d/h$ is the head dimension for $h$ heads. Each head computes scaled dot-product attention with ALiBi biases $B$:

$$\text{Attention}(Q, K, V) = \text{softmax}\left(\frac{QK^\top + B}{\sqrt{d_h}}\right) V,$$

where $\text{softmax}\left(\frac{QK^\top + B}{\sqrt{d_h}}\right) \in \mathbb{R}^{\ell \times \ell}$ defines the attention weights.

After model training (on the full dataset using the default hyperparameters identified via cross-validation), we pass in the full training data to compute these attention weights. For each sample, we extract the self-attention matrix from the final transformer layer for each head. We can average these matrices across all samples and all heads to obtain a global attention score matrix:

$$\bar{A} = \frac{1}{hN} \sum_{n=1}^{N} \sum_{j=1}^{h} A_n^{(j)},$$

where $A_n^{(j)} \in \mathbb{R}^{\ell \times \ell}$ is the attention matrix for the $j$-th head on the $n$-th region, and $N$ is the total number of regions.

For subnetwork-level analysis, we instead compute attention weights restricted to samples belonging to canonical functional subnetworks defined by our UK Biobank 456 region, 9 network parcellation. We compute the attention weights for each network $k \in \{1, \ldots, 9\}$ by passing in its regions to the encoder individually. These attention matrices are aggregated in the same way, producing a network-specific average attention map $\bar{A}_k$. This procedure allows us to characterize how attention is distributed over gene tokens in general, and emphasize distinct transcriptomic features attended to for different brain networks.

Figure 13A shows that, in the absence of the `[CLS]` token, self-attention patterns differ across heads, with specific gene tokens receiving higher attention in some heads than others. These variations also exhibit different spatial frequencies, possibly reflecting local versus global transcriptomic interactions. Such diversity may be encouraged by the ALiBi slope mechanism.

Figure 13B presents the mean attention scores for the SMT w/ `[CLS]` model. The first token (expanded for visibility) is the `[CLS]` token, which consistently receives the highest attention across all heads, indicating strong influence from spatial coordinates. Interestingly, the 54[th] gene token (located on chromosome 8) receives relatively high attention scores in all heads, suggesting that despite the strong emphasis on spatial position, some gene modules such as those related to white matter microstructure, neurodevelopment, or metabolic activity, may selectively co-embed with spatial information to improve prediction. The stability and significance of these attention heads must be further validated to link the gene sets highlighted by the SMT with biologically relevant processes.

Figure 13C shows average attention distributions when regions from specific functional subnetworks are separately input into the SMT. Across a subset of six functional networks, we observe the subcortex and cerebellum show similar profiles to each other as compared to cortical networks. Certain gene tokens consistently receive elevated attention across all subnetworks. Further analysis of global and subnetwork-specific attention patterns may help determine whether distinct transcriptomic modules underlie functional connectivity across the brain.

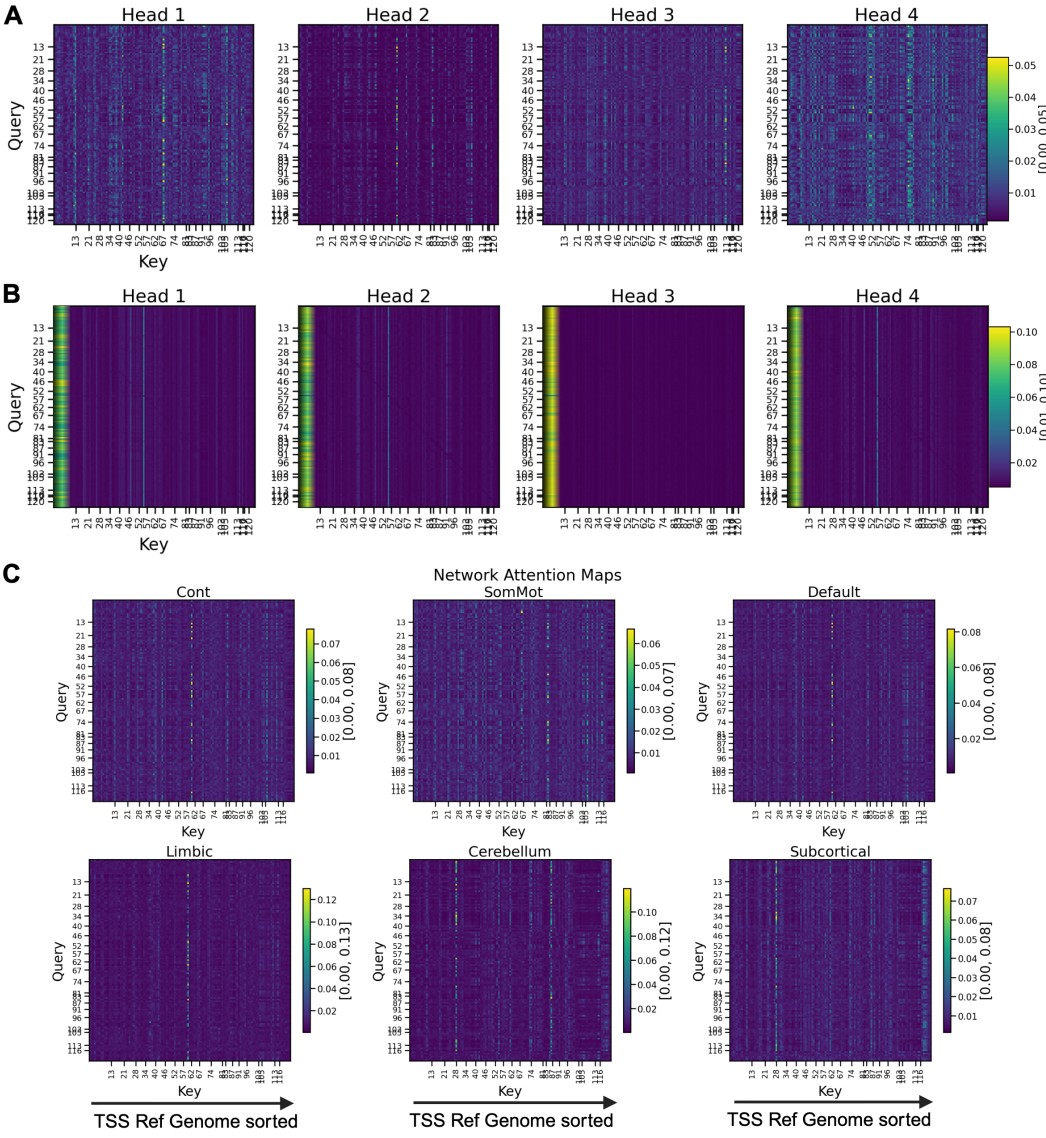

Figure 13: **SMT multi-head self-attention weights. [A]** Attention scores for the Spatiomolecular Transformer. Tick marks on both Query and Key axes denote token positions at chromosome boundaries. **[B]** Attention scores for SMT with [CLS] token. The [CLS] token key column is artificially expanded for visualization. **[C]** Subnetwork-specific attention weights for SMT, averaged over all heads for a subset of six functional subnetworks.

Table 9: Author contributions.

| | AR | SG | JW | CD | EV |
|---|---|---|---|---|---|
| Study concept or design | ■ | | ■ | ■ | ■ |
| Data acquisition or processing | ■ | | ■ | ■ | |
| Data analysis or interpretation | ■ | ■ | | | |
| Drafting/revision of manuscript for content | ■ | ■ | | | ■ |
| Funding acquisition | ■ | | | ■ | ■ |

**Note:** Bolded initials indicate the primary contributor. Black cells indicate a documented contribution to the corresponding activity.

