# OpenReview forum: "Predicting Functional Brain Connectivity with Context-Aware Deep Neural Networks"
_NeurIPS.cc/2025/Conference — NeurIPS 2025 poster_

### Official Review · Reviewer_a6QV · 2025-07-02

**Clarity:** 2
**Significance:** 3
**Originality:** 3
**Rating:** 4
**Confidence:** 3

**Summary:**

This paper proposes using a transformer model SMT to predict brain connectivity from high-dimensional transcriptomic data, incorporating coordinate information to capture each brain region's 3D position. For evaluation, the authors leverage a Spatiomolecular Null Mapping technique to help rigorously control for spatial autocorrelation effects. Across within-dataset tasks on UK Biobank and MPI-LEMON, as well as cross-dataset and cross-resolution tasks, SMT consistently outperforms baseline methods. The interpretability analyses presented in the supplementary material further elucidate the mechanisms by which SMT makes its predictions.

**Questions:**

1. Why is the UMAP analysis for the MLP not provided in Figure 8?

2. In the supplementary material (line 565), Figure 4 does not seem to include results for SMT w/ [CLS].

**Ethical Concerns:**

["NO or VERY MINOR ethics concerns only"]

**Final Justification:**

This paper introduces a nonlinear model that maps gene expression to functional connectivity, with evaluations conducted on two large-scale datasets (UKB and MPI-LEMON). The methodological clarity, well-structured manuscript, and great design of spatiomolecular null mapping baselines and the corresponding analysis are strong aspects of the work.

In the rebuttal, the authors have addressed several previously raised concerns. Notably:

- They clarified the processing pipelines for the datasets.

- They provided additional details regarding the architectural configuration of SMT and MLP.

- They elaborated on the procedure for projecting Euclidean coordinates using fixed layers.

These clarifications enhance the clarity and presentation of the paper. However, several important concerns remain unresolved:

**Major concern:**
The central issue lies in the unclear benefit of using a Transformer architecture (SMT). The SMT model does not demonstrate a clear performance advantage over the MLP baseline, yet it introduces greater architectural complexity. As currently presented, the use of a Transformer appears unnecessary, and the model lacks architectural improvements that would justify its use in this context. This raises concerns about the clarity of the model’s contribution, and the inclusion of “Transformer” in the title may overstate its relevance.

**Additional unresolved (minor) issues:**

- The rebuttal does not clearly explain the selection or scheduling of the initial augmentation probability.

- The information provided on hyperparameter tuning remains insufficient. The authors mention grid search but do not specify which parameters were tuned and what candidate values were explored.

In summary, the paper presents an interesting nonlinear modeling framework with comprehensive evaluations. However, the necessity and added value of using a Transformer architecture remain insufficiently justified, which affects the clarity and focus of the paper’s contribution.

**Limitations:**

The author has adequately discussed the limitations.

**Paper Formatting Concerns:**

No formatting issues found.

**Quality:**

3

**Strengths And Weaknesses:**

**Strengths**

- The paper introduces a nonlinear model mapping gene expression to functional connectivity. Comprehensive evaluations on two UKB and MPI-LEMON datasets demonstrate that the proposed nonlinear models can better capture the robust relationship between gene expression and functional connectivity compared to baseline approaches.

- The design of the baselines is great. The analysis based on rigorously defined spatiomolecular null mapping baselines clearly disentangles the respective roles of spatial location and gene expression in predicting functional connectivity.

- The methodology is clearly described, and the overall manuscript is well-structured.

**Weaknesses**

- The major concern is that the proposed SMT does not appear to significantly outperform the MLP model. This raises questions about the necessity of adopting a more complex transformer architecture. The same types of analyses, such as UMAP and activated gene segments, could also potentially be performed using an MLP. The paper does not clearly articulate the specific advantages of SMT or the additional insights gained from using it.

- Other concerns relates to clarity in implementation details:

    1. The process of constructing the population-averaged paired dataset from the Allen Human Brain Atlas and UKB needs to be clearly described. Additionally, details on the amount of data remaining after population-averaging should be provided to improve understanding and reproducibility.

    2. The setting and the impact of $\alpha_0$ in the Target-Side Distributional Augmentation needs clarification.

    3. In Table 6, how exactly were the Euclidean coordinates projected through fixed layers implemented?

    4. The architecture details of SMT and MLP should be described. Furthermore, the hyperparameter search ranges used in the grid search should be reported to enhance reproducibility.

---

> ### Author Rebuttal · Authors · 2025-07-31
>
> We thank reviewer a6QV for their helpful and constructive feedback. In response, we have clarified the contributions of our study and expanded on several details, including population-averaging strategies and architecture components. We also thank the reviewer for their appreciation of our baseline and null modeling approach. Many of the baseline models we apply, such as the Connectome Model [1] and its low-rank form [2], have not yet been benchmarked on human data. The reviewer makes an excellent point regarding the comparable performance of SMT and MLP. We view this as a central finding of our study that modern non-linear architectures offer a significant leap in performance over traditional linear models, of which we believe this is the first paper this is studied and observed.
>
> We hope to present both models not as competitors, but as exemplars of two distinct deep learning approaches in this problem space. The MLP establishes a powerful, general-purpose baseline. In contrast, the SMT provides a modular framework specifically designed to leverage the sequential nature of genomic data and test biologically-informed priors. This architecture enables unique advantages in 3 ways 1) its modularity for incorporating context, 2) its ability to take advantage of pretrained weights from other genomic foundation models, and 3) its interpretability via analyzing attention weights. We have revised the Discussion to articulate this framework and expanded our qualitative analysis to demonstrate the unique scientific insights afforded by the SMT described below:
> 1. **Modular components.** The SMT is introduced as a unified model for connectome prediction, with modular components that integrate biological features such as genomic position and spatial context. The transformer architecture is well-suited for modeling structured, semantically rich inputs like genetic sequences, enabling more targeted hypothesis testing—such as evaluating the impact of transcription start site (TSS) ordering on connectivity prediction [3]. It also facilitates richer integration of auxiliary data beyond simple concatenation, using context tokens (e.g., [CLS]/[SEP] in BERT [3]) to encode chromosomal boundaries or functional gene groupings. This modularity makes SMT a strong foundation for future extensions incorporating personalized demographic or genomic information via learned vectors or embeddings.
> 2. **Performance.**
> We believe that incorporating pretrained models with finetuning may yield substantial improvements when paired with components described above. This is supported by findings across genomic foundation models such as HyenaDNA [4] and scBERT [5] citing large performance gains with pretraining. We discussed why this strategy was not yet implemented with reviewer NCYX, but which we will clarify in Future Directions.
> 3. **Interpretability.**
> We present an extension of our Figure 9 results to reviewer gLbH, demonstrating how attention weights can be paired with Gene Ontology tools to further scientific insight. This procedure enabled us to identify two relevant neuron projection gene modules that may be mechanistically related to functional connectivity.
> SMT attention offers a potential mechanistic bridge between GWAS loci and brain connectivity by highlighting not only which genomic regions are important, but also how they interact transcriptionally. For example, 19p13.11, an autism-associated region from Simons Foundation Autism Research Initiative maps to token 106 in our TSS-sorted input, which receives consistently high attention in cortical regions (see reviewer gLbH). The combination of token-level weights and token-token interactions are unique to attention-based models and may be difficult to extract using MLPs.
>
> ## Implementation details
> Here, we add further details to be added to Section 2.1 and Supplement A2 to clarify the population averaging steps for the Allen Human Brain Atlas (AHBA), UKBB, and MPI-LEMON datasets.
>
> ### Gene expression processing
> 1. **Within-Donor Normalization.**
> Gene expression values for all tissue samples are normalized within each donor to account for individual differences in signal scale or distribution. This is done using a scaled robust sigmoid transform, which standardizes values while preserving their rank order:
> $$
> x_{\text{norm}} = \frac{1}{1 + \exp\left( -\frac{x_g - \text{median}(x_g)}{\text{IQR}(x_g)} \right)}
> $$
> Here, x_g is the expression vector for gene g across all brain regions in a given donor, median(x_g) is its median, and IQR(x_g) is its interquartile range. This transformation is followed by min-max rescaling:
> $$
> x_{\text{scaled}} = \frac{x_{\text{norm}} - \min(x_{\text{norm}})}{\max(x_{\text{norm}}) - \min(x_{\text{norm}})}
> $$
> Unlike z-scoring, this two-step normalization procedure is distribution-free and robust to outliers, allowing expression values to be placed on a comparable scale across donors.
> 2. **Sample-to-Region Assignment.**
> For each subject, tissue samples are assigned to the nearest brain region in the target parcellation using MNI coordinates, with anatomical constraints on hemisphere and cortical vs. subcortical divisions. Expression values are averaged within each region, yielding an r × g matrix (regions × genes) per subject.
> 3. *Gene Selection and Filtering.*
> To ensure that downstream analyses focus on biologically reliable signals, we apply a multi-step gene filtering procedure designed to retain only genes with consistent spatial expression patterns across donors. First, genes with expression below background noise are excluded. Then, we compute the differential stability of each gene g as:
> $$
> \Delta_S(g) = \frac{1}{\binom{N}{2}} \sum_{i=1}^{N-1} \sum_{j=i+1}^{N} r(B_i(g), B_j(g))
> $$
> Where B_i(g) is the regional expression profile of gene g in donor i, and N = 6 is the number of donors. Genes with high Δ_S(g) exhibit stable, reproducible spatial variation. We apply the strictest threshold recommended for the Schaefer-400 parcellation, yielding 7,380 stable genes.
> 4. **Cross-Donor Aggregation**
> After normalization and filtering, expression values are averaged across donors. Missing regions are filled via bi-hemispheric mirroring and Gaussian smoothing over the 10 nearest neighbors, producing a complete region-by-gene matrix aligned to the target parcellation.
>
> ### Functional connectivity processing
> Each subject underwent a resting-state fMRI scan capturing low-frequency BOLD fluctuations across brain regions. Functional connectivity (FC) was computed using Pearson correlation between the BOLD time series of each region pair, yielding an r x r FC matrix per subject. These matrices were then averaged across 1814 UKBB participants to produce a population-level FC matrix representing typical patterns of coordinated brain activity. The UKBB connectome is parcellated into 456 regions (~103k connections). As shown in Fig. 3A, the matrix exhibits canonical subnetwork structure. A similar connectome from 1065 HCP-YA (young adult) subjects (Pearson r = 0.9) suggests a stable population-level architecture across cohorts despite demographic differences.
>
> ### Target augmentation
> Target-side distributional augmentation was introduced to address the scarcity of unique transcriptomic samples relative to the abundance of connectome data, inspired by principles of curriculum learning/scheduled sampling. The strategy involves replacing target connectivity values with realistic alternatives drawn from the population distribution, intended as an ad-hoc pretraining/regularization strategy. As shown in Fig 4 and 7, the augmentation provides a modest performance gain across test and validation splits. While the effect is limited in the current setup, we believe this protocol offers a promising foundation for future work, particularly when combined with distribution-based loss functions or alternative sampling schedules, and may be useful for others tackling data imbalanced biological prediction problems.
>
> ### Coordinate linear projection
> In Figure 6, the Euclidean coordinates are projected through a linear layer into the same embedding space as the gene tokens, functioning as a CLS token that participates in attention. The projection weights are initialized from a uniform distribution between –k and k, where k is equal to 1 divided by the number of input features; in the learned case, the layer is differentiable, while in the fixed case, the weights remain frozen throughout training.
>
> ### SMT/MLP details
> Details on hyperparameter selection and search ranges are provided in Section A.2 of the supplement, and the exact grids for each model are available in the repository alongside model code in the /models directory. The MLP used in this study is a 2/3 layer architecture with dropout, layer norm, and AdamW optimizer trained on MSE loss, with learning rate scheduling and early stopping. The SMT uses a 2/4-layer transformer encoder with 2/4 attention heads and a similar MLP decoder; its token-based input reduces parameter count and encourages biologically meaningful genomic interactions.
>
> ### Response to Questions
> 1. The UMAP analysis for the MLP was omitted due to space limitations and because the MLP lacks a dedicated encoder, however, embeddings could be extracted from its final layer. We can include this analysis in the revised version for completeness.
> 2. The results for SMT w/ CLS are present in Table 5.
>
> References
> 1. Kovács, I. A. et al. Uncovering the genetic blueprint of the C. elegans nervous system. 2020.
> 2. Qiao, M. Deciphering the genetic code of neuronal type connectivity through bilinear modeling. 2024
> 3. Nguyen, E. et al. HyenaDNA: Long-range genomic sequence modeling at single nucleotide resolution. 2023
> 4. Devlin, J. et al. BERT: Pre-training of Deep Bidirectional Transformers for Language Understanding, 2019
> 5. Yang, F. et al. scBERT as a large-scale pretrained deep language model for cell type annotation of single-cell RNA-seq data. 2022

---

> ### Comment · Reviewer_a6QV · 2025-08-06
>
> Thank the authors for the detailed response. Based on the response, several concerns remain as outlined below:
>
> **Major concern:**
> While SMT may have potential for extensions in future work, the current application of the Transformer lacks clear motivation or justification, especially since it is used without architectural modifications or innovative design. If the Transformer performs similarly to an MLP, the necessity of using a Transformer becomes questionable. This further raises the issue that the use of “Transformer” in the paper’s title may be misleading, as the core contribution seems to lie more in the non-linear modeling aspect rather than in any Transformer-specific advancement.
>
> **Minor concerns** regarding clarity and presentation:
>
> If my understanding is correct, the paper and supplementary material do not clearly explain how the initial augmentation probability is selected. Supplementary Material A.4 mentions a 0.45 augmentation rate — does this correspond to the initial augmentation probability, or to the final probability after scheduling? This should be clarified.
>
> The response regarding the hyperparameter search range in Supplementary A.2 remains vague. It would be helpful to clearly specify which parameters were optimized in hyperparameter tuning and provide the candidate values. A table summarizing this information would improve clarity.

---

> ### Author Response · Authors · 2025-08-07
> **response to concern 1**
>
> We thank reviewer a6QV for their continued engagement in our research and helpful suggestions. We understand the concern that the core contribution of our paper may lie in the introduction of non-linear modeling for this novel scientific problem, and our framing and title should better reflect this.
>
> To better align the paper's narrative with its central findings, we will revise the abstract and main contributions in the introduction to emphasize the conclusive predictive power of non-linear architectures (both MLP and SMT) against linear baselines and our stringent null model. Below is our revised abstract excerpt:
>
> “... This paper presents a first of its kind, comprehensive evaluation of multimodal modeling strategies for predicting whole-brain functional connectivity from gene expression and regional spatial coordinates. We contrast feature-based and existing bilinear models with deep neural network approaches, including our proposed attention-based Spatiomolecular Transformer (SMT) and a comparable MLP-based architecture, to demonstrate the superior generalization ability of non-linear methods beyond chance. … (SMT and SM null map sentences)... Across multiple connectomics datasets (UK Biobank and MPI-LEMON) and varying parcellation resolutions, we find that multimodal non-linear models achieve superior predictive performance, especially for long-range connections, with predictions significantly exceeding what is expected from spatial autocorrelation…”
>
> Below is our revised main contributions:
> - We introduce the first deep learning approaches for predicting functional connectivity from regional gene expression in humans, featuring an attention-based architecture (SMT) with biologically informed, modular components that enable targeted ablations and hypothesis testing.
> - We introduce and validate a novel spatiomolecular null mapping technique that generates surrogate gene expression maps preserving not only spatial autocorrelation but also key  transcriptomic correlation structures, providing a highly stringent benchmark for assessing genuine predictive signal. We use this to assess the significance of both our novel non-linear architectures but also existing linear methods.
> - Across several train-test split settings, non-linear methods including MLP and SMT, achieve superior performance in predicting functional connectivity to unseen brain regions. This performance significantly exceeds that of our rigorous null models and is shown to generalize across multiple connectomic datasets (UK Biobank, MPI-LEMON) and varying parcellation resolutions.
>
> We believe these changes will make the paper's contributions clearer and more impactful. Our goal is to shift the focus from a single model to the broader framework, while still capturing the unique biological context we incorporate. Regarding the title, we understand our suggestion may have been too broad and offer an alternative to clarify our contributions: "**Predicting Functional Brain Connectivity with Context-Aware Deep Neural Networks**", in the case we are able to update it. If you find that these changes improve the clarity and impact of our paper, we would be very grateful if you would consider updating your review and score.

---

> ### Author Response · Authors · 2025-08-07
> **response to concern 2**
>
> Thanks again for the reviewer's suggestions to help improve the clarity of our study. To address the minor concern, all model code and hyperparameter grids can be found in our repo linked in the abstract under _/models_ and _/models/configs_ respectively. We agree that including a summary table would improve clarity, therefore, we include a table below for the SMT w/ [CLS] model’s hyperparameter grid to be added to the main paper.
>
> To select the optimal hyperparameters, for each fold of the 4-fold CV split, the dataset is split into train, validation, and test sets. 5 random configurations are sampled from the grid. The configuration with the lowest validation loss (MSE) is used for final model fitting on the training folds.
>
> | Hyperparameter        | Values                          |
> |-----------------------|----------------------------------|
> | token_encoder_dim     | [20, 60]                         |
> | d_model               | [128]                            |
> | encoder_output_dim    | [10]                             |
> | use_alibi             | [True, False]                    |
> | nhead                 | [2, 4]                           |
> | num_layers            | [2, 4]                           |
> | deep_hidden_dims      | [[256, 128], [512,256,128]]                     |
> | cls_init              | ['spatial_learned']              |
> | transformer_dropout   | [0, 0.2]                         |
> | dropout_rate          | [0, 0.2]                         |
> | learning_rate         | [0.00009]                        |
> | weight_decay          | [0.0001]                         |
> | batch_size            | [512]                            |
> | aug_prob              | [0, 0.1, 0.3, 0.45]              |
> | epochs                | [50, 70]                         |
>
> Some hyperparameters are fixed and others have a range. ‘aug_prob’ defines the target augmentation probabilities, which starts with a selected probability and decays linearly. Thus, if the selected aug_prob value is 0.45, then with 0.45 chance the targets are augmented at epoch 1, with 0.225 they are augmented at epoch 35, and at epoch 70 the probability of augmentation is 0. The decay rate is linear regardless of the starting value. This strategy was introduced to encourage early exposure to subject-level variability while gradually guiding the model toward the population structure.
>
> For the main results in Table 1, we followed the cross-validation strategy, sampling from the grid above. For the target augmentation ablation experiment in Figure 6, the target augmentation probabilities are grid searched with all other components ablated. For example, in the top right panel of Figure 6, we show how target aug prob influences performance when CLS and Alibi slopes are not included. The ‘final’ value of 0.45 refers to the starting probability used in Figure 4. Here we test the components more additively, starting with the base SMT, then adding in target augmentation with the maximal augmentation schedule, followed by AliBi slopes and the spatial CLS token.
>
> These details will be clarified in the Hyperparameter Experiments section of the Supplementary Material. Several ablation experiments were introduced, including a new experiment described to reviewer NCYX, to fairly assess the influence of the modular components of the SMT and interactions among the hyperparameters.

---

> > ### Comment · Reviewer_a6QV · 2025-08-08
> >
> > Thank the authors for the reply. I have no further questions.

---

### Official Review · Reviewer_gLbH · 2025-07-05

**Clarity:** 3
**Significance:** 2
**Originality:** 2
**Rating:** 5
**Confidence:** 4

**Summary:**

The manuscript entitled "Predicting Functional Brain Connectivity with a Context-Aware Spatiomolecular Transformer" presents a transformer-based model (Spatiomolecular Transformer (SMT)) with biologically-grounded data augmentation techniques to map functional brain connectivity with spatial transcriptomic data.

The model tokenizes gene expression by transcription start site (TSS) to preserve genomic context and includes a spatially initialized [CLS] token to account for brain region location. The proposed SMT model is compared with several linear and non-linear baselines models using datasets from UK Biobank and MPI-LEMON at various parcellation scales. Results demonstrates superior performance of SMT, especially in predicting long-range connections, indicating its capability to capture real biological mechanisms rather than just local similarities.

**Questions:**

The reviewer is curious why the author formulate the problem only as a "prediction", rather than a "mapping" task, which has been extensively emphasized but never explained. What will be the limit to do the two-way mapping between functional connectivity and transcriptomic data?

**Ethical Concerns:**

["NO or VERY MINOR ethics concerns only"]

**Final Justification:**

The authors provided additional neuroscientific analyses of attention weights that are interesting and help contextualize the results, but these additions were not part of the original manuscript and only partly address the concern that the paper does not sufficiently leverage its method for deeper neuroscientific insight. Methodologically, while the author argued that SMT framework is competently implemented and shows some gains, its performance is only slightly better than a standard MLP and the novelty of the architecture remains limited. For these reasons, I retain my original rating.

**Limitations:**

In addition to the more general comments listed above in the "Summary", the model’s generalization from UK Biobank to MPI-LEMON (Pearson r = 0.55) is relatively weak given the high-dimensional nature of the input. While some variation is expected due to demographic and methodological differences, the paper doesn't fully account for this or examine other transfer learning strategies to improve robustness.

**Quality:**

3

**Strengths And Weaknesses:**

The main task of this paper: mapping fMRI-derived functional connectivity from spatial transcriptomics, is an important task to bridge microscope, molecular and macroscope, systems-level neuroscience. However, the authors did not seize this opportunity to investigate deeper neuroscientific insights from the learned mappings. Considering the space limitation of a NIPS submission, the lack of such investigation is understandable, but definitely shall be work on in its future iterations. For example, gene sets contributing to long-range vs. short-range connectivity, or molecular signatures of subnetworks (e.g., default mode vs. visual), are only briefly discussed or left in the appendix. Recent neuroscience works pointed out the utility of spatial transcriptomics for uncovering mechanisms of cortical specialization, which this paper could have explored in more depth.

Methodologically, although the model structure is not that complicated, SMT can advance the field by providing more effective, multimodal integration of gene expression and brain spatial structure, which is much needed in system neuroscience. Additionally, SMT’s architecture allows for interpretable attention maps, which could potentially reveal gene pathways underlying connectivity phenotypes. The development of distributional augmentation sampling subject-level fMRI matrices during training is a creative solution to the low sample size of transcriptomic data. It brings in inter-individual variability from fMRI while maintaining a fixed transcriptome. While the augmentation benefits were modest in ablation studies, this technique is of technical merit and aligns with trends in data-rich domains like genomics. It could serve as a template for integrating population data in low-sample transcriptomic domains in a more general perspective.

---

> ### Author Rebuttal · Authors · 2025-07-31
>
> We thank reviewer gLbH for their supportive comments of our manuscript and appreciate the understanding that given the space limitations, the manuscript is primarily focused on experimental design and modeling results. While we did perform some high-level evaluation of attention weights in Figure 9, we are inspired by the reviewer comments and have extended this analysis further to bring deeper neuroscientific insight to our study.
>
> ## Neuroscientific insight
> In Figure 9C, the average attention weights for 6 of the 9 functional subnetworks are visualized. The tokens are sorted by chromosomal position, with 23 chromosomal boundaries denoted by tick marks on the axes. To gain additional insight from the attention weights, we have performed column-wise sums for each subnetwork resulting in one vector per subnetwork that denotes the total attention ‘paid’ to each token. This set of 9 vectors can be compared by subnetwork to identify token-wise similarities or differences. A visual representation will be included in Figure 9.
>
> As can be observed in Figure 9C, each functional subnetwork attention matrix tends to have 1-3 tokens with stronger attention weights. Using the summing procedure described above, we **display the top 2 tokens attended to for each functional subnetwork**:
> - Visual: 106, 82
> - Somatomotor: 82, 106
> - Dorsal Attention: 106, 38
> - Salience Ventral Attention: 60, 42
> - Control/Frontoparietal: 60, 106
> - Default: 60, 106
> - Limbic: 60, 112
> - Subcortical: 28, 118
> - Cerebellum: 28, 86
>
> A clear pattern arises. Across the cortex there are 3 common tokens attended to - token 60, 82, and 106. Across the subcortex there is a single common token - token 28. Given the token size of 60 we now have a set of 180 genes for the cortex and 60 for the subcortex to analyze further.
>
> Although sorted by TSS, it is possible that some of the genes in these groups are unrelated to the transcriptome-connectome subspace, which may introduce undesirable noise for Gene Ontology analysis. Therefore, we use **HumanBase, which applies tissue specific community detection to find cohesive gene clusters from a provided gene list**. The module detection enables systematic association of gene groups related to specific tissues or processes. Instead of processing the full gene group, resulting modules are tested for functional enrichment of Gene Ontology biological process terms.
>
> Of the 180 genes in the top 3 cortex tokens, three distinct modules are detected. **The first cortex module M1 contains 35 genes enriched for positive regulation of dendrite extension. Of the 60 genes in the subcortex token, the first module M1, contains 15 genes enriched for negative regulation of neuron projection development.** Modules only contain terms passing significance thresholds using one-sided Fisher’s exact tests and Benjamini–Hochberg corrections to correct for multiple tests. HumanBase hyperlinks with full gene lists and enrichment results will be included in the final manuscript as links are not permitted in the rebuttal phase.
>
> ### *Interpretation:*
> Since functional connectivity reflects coordinated hemodynamic activity, one may expect enrichment of gene pathways related to neuronal activity and metabolism. Instead, we identified two modules associated with the regulation of selective neuronal wiring patterns - an upregulating module in the cortex and downregulating module in the subcortex. This suggests that the genetic gradients shaping structural wiring during development may also support the maintenance of functional circuits in adulthood. These findings align with Vogel et al.’s [1] observation that developmental transcriptomic gradients persist into adulthood. Moreover, the emergence of neuronal projection modules in the attention weights of the SMT echoes results from several mouse studies , which show that genes linked to neuronal connectivity often correlate with long-range connectivity patterns [2]. These results will be clearly added to our main manuscript and supplement.
>
> We hope to expand on these findings in future work by generating null distributions of attention weights and evaluating attention across subpopulations as an avenue for deriving imaging-genetics phenotypes. In summary, the inclusion of clearer attention weight visualization, paired with principled Gene Ontology analysis serves to show how the SMT can contribute to neuroscientific insight regarding the transcriptome-connectome relationship.
>
> ## Novelty of Target Augmentation
> We appreciate the comment regarding the novelty of our target-side augmentation protocol and its potential as a template for future work. This strategy leverages a key asymmetry in the field: the scarcity of high-quality gene expression data versus the abundance of connectivity data. One direction we are actively exploring is joint feature-side and target-side augmentation, where each time a new target connectivity value is sampled, the corresponding transcriptomic input is perturbed as well. Specifically, we introduce small amounts of noise to the gene expression vectors, scaled by the variance of each gene across the six AHBA donors. Pairing true connectivity values with realistically perturbed transcriptomic inputs may serve as a form of regularization, potentially improving generalization. We are currently testing this approach along with additional strategies inspired by the reviewers’ suggestions—such as selective sampling, a common technique in imbalanced classification settings. We look forward to reporting these future directions and incorporating them in the next iterations of the SMT.
>
> ## Prediction vs Mapping
> We frame our study as a prediction instead of a mapping task because (1) we have a single population average transcriptomic map and (2) we want to test more complex deep learning based methods on this task.
>
> If paired transcriptomes and connectomes were available for multiple individuals, the problem could be framed as a true mapping task by holding out entire subject-level pairs for testing. Since we only have population averages here, we instead choose to partition connections of the brain into training and testing sets to evaluate whether learned transcriptome-connectome patterns generalize. If framed as a mapping task using the full population-average data, even a simple MLP with sufficient parameters could trivially overfit the connectome, whereas more constrained models like the Bilinear Connectome Model (see Figure 5) are not expressive enough to fully capture the variance.
>
> We are excited to explore this problem as a mapping task in future work, potentially with graph-based methods, as more paired imaging-genetics are collected.
>
> ## Transfer Learning
> In most cases the cross-dataset generalization exceeds r>0.7, but the reviewer rightly points out that when trained on MPI 183 resolution, the model only reconstructs UKBB with a pearson-r of 0.55. While we do state that demographic differences may contribute to this result, this outlier result is much more likely due to parcellation differences. The MPI-183 parcellation has ~16k unique connections to train on whereas UKBB-S456 has ~103k. With far fewer samples in a more coarse spatial resolution, it is understandable that the MPI-183 model struggles to generalize across datasets and even within-dataset to more resolved parcellations.
>
> We agree that additional transfer learning strategies could be further explored. One potential avenue that we are open to discussing with the reviewers, would be to add a metadata token to the model, so that a unified model can be trained across all datasets with awareness to demographics like age, sex, and ethnic background. This could lead to a more robust model and exploration of differences in subpopulations.
>
> References
> 1. Vogel, J. W. et al. A molecular gradient along the longitudinal axis of the human hippocampus informs large-scale behavioral systems. 2020
> 2. French, L. et al. Large-scale analysis of gene expression and connectivity in the rodent brain: insights through data integration. 2011

---

> > ### Comment · Reviewer_gLbH · 2025-08-06
> >
> > The reviewer read through the comments from other reviewers and the author's response. The author's investigation in "Neuroscientific insight" is interesting and cast some insight into the experiment results, yet is not part of (and will not likely to be included) in the paper. Thus the reviewer is more concerning about the lack of novelty in its methodology design as raised by the other reviewers, and the relatively sub-par performance (slightly better than MLP) of the proposed model. It seems that there is could be potentially a better, more carefully implemented model for this task. With the considerations above, the reviewer will retain the original evaluation on the quality of this work.

---

> ### Author Response · Authors · 2025-08-06
>
> We appreciate the insightful remarks from reviewer gLbH and for their continued support of our manuscript. We agree that the Neuroscientific Insight analysis outlined above adds insight to our experimental results. Although this analysis is not part of the original manuscript, it is included in our final version, discussing our insights in the Results section and adding an additional panel to Figure 9D.
>
> We hope that our explanation clearly outlines how attention weights can be used to identify chromosomally contiguous gene groups that receive high amounts of attention, and how this information can inform our understanding of functional subnetworks. While we cannot include the figure in our response here, a comparable figure for reference can be found in Figure 4A of CellSpliceNet [1]. We believe this analysis to be crucial for demonstrating gene token interpretability and to situate our results with prior findings in the field. Finally, we align with the reviewer’s comment that the SMT can be improved further by using the current components as a template for future investigation.
>
> 1. Afrasiyabi, A. et al. CellSpliceNet: Interpretable multimodal modeling of alternative splicing across neurons in C. elegans. 2025

---

### Official Review · Reviewer_NCYX · 2025-07-06

**Clarity:** 4
**Significance:** 3
**Originality:** 3
**Rating:** 4
**Confidence:** 3

**Summary:**

This paper introduces the Spatiomolecular Transformer (SMT), an attention-based architecture designed to predict human brain functional connectivity from gene expression data and spatial coordinates. The work addresses a fundamental question in neuroscience: can we predict how brain regions connect based on single-cell spatial transcriptomics. This work does not focus on improving single-cell representation and/or imputation but rather aims at leveraging recent prior works on single-cells and to apply on a real-world application.

The Spatiomolecular Transformer (SMT) tokenizes genes based on their genomic position (transcription start sites) and uses multi-head self-attention with ALiBi biases to capture both local and global genetic interactions. The authors incorporates 3D brain coordinates through a dedicated [CLS] token that participates in the attention mechanism, allowing spatial information to modulate molecular features. One key experiment in this work is also the Spatiomolecular Null Mapping, to verify that the SMT architecture learn more than simply spatial correlations.

**Questions:**

- Contributions: are the authors the first ones to apply a Transformer's architecture to predicting functional connectivity ? This could be clearly stated in the contribution as well.
- TSS ranking: if you do not use the TSS ranking but rather a random split, what is the impact in terms of performance ? It would be interesting to add this in the ablation study.
- Architecture wise: what is the impact of the learned linear map ? Could it be replaced by 1D convolutions ? What is the impact of the Transformer block vs only 1D convolutions ?
- Pre-training: what would be the benefits of leveraging a checkpoint pre-trained on dozens of millions of single-cell RNA sequences ?
- Number of genes: why did the authors limit themselves to 7380 genes ?
- The loss is not discussed in the main paper nor in the Appendix. I believe could also yield to interesting discussions. I may have missed something but, similarly, I believe different sampling schemes could be discussed for example to ensure balanced representation of short/medium/long-range connections. Do you think this could be useful in your setting ?

**Ethical Concerns:**

["NO or VERY MINOR ethics concerns only"]

**Limitations:**

One of the two main limitations of this paper, in my opinion, are the following:
- clearly stating whether it is the first application of Transformers for the task of Brain Connectivity (if not, adding some baselines would greatly improve the comparison).
- as mentioned the questions above, extending the ablation study would help understand better what elements in this SMT architecture brings performance.

**Paper Formatting Concerns:**

No formatting concerns.

**Quality:**

3

**Strengths And Weaknesses:**

**Strengths**

- The problem is extremely relevant: the authors address a fundamental question linking molecular biology to brain function with clear neuroscientific relevance.

- A biologically relevant preprocessing and modeling: "sorting each region’s expression vector by the genomic position of genes based on their transcription start site" this is a witty and smart way to rank genes that has biological meaning. It assumes that genes that are physically close on chromosomes often work together or are regulated together. Also, the ALiBi attention slopes incorporate meaningful biological priors about genomic organization.

- Causality: The spatiomolecular null testing is particularly well-designed, addressing a critical confound (spatial autocorrelation) often overlooked in neurogenomics studies. This experiment clearly underlines what the SMT architecture learns apart from spatial correlation.

- Comprehensive Evaluation: Tests across multiple datasets (UK Biobank, MPI-LEMON) and parcellation resolutions, with proper cross-validation and distance-stratified analyses.


**Weaknesses**

- Limited Novelty: While well-executed, the transformer application is relatively straightforward. The performance gains over simpler MLP baselines are modest (SMT: 0.79 vs MLP: 0.78 Pearson-r). The authors briefly state that "the attention mechanism does not substantially outperform comparable non-linear baselines", this raises questions about the added complexity's value. "It does enable key design features for qualitative evaluation": this qualitative study could have been extended, the UMAP projections do not illustrate a qualitatively superior embedding space for SMT.
- I believe that the SMT architecture could have been studied slightly more, meaning a larger ablation study. What is the impact of the learned linear map ? Could it be replaced by 1D convolutions ? What is the impact of the Transformer block vs only 1D convolutions ?
- Also, if there is "a scarcity of human brain gene expression data", there are hundreds of millions of single-cell data in open public datasets. It could have been interesting to measure the impact of a pre-training on a large non-brain dataset. Beside, there are already some existing pre-trained checkpoints available (scBERT, scGPT, cellPLM, sCT...).
- It seems that the authors limit themselves to 7380 genes, why is that ? how is the choice done ? could we maybe increase this to reduce the assumptions done ?

---

> ### Author Rebuttal · Authors · 2025-07-31
>
> We thank reviewer NCYX for their insight and helpful suggestions, which has prompted us to clarify our experimental contributions, expand our ablation study, and evaluate alternative model features.
>
> ## Novel contributions
> - To address the reviewer’s first question, to the best of our knowledge, **this study presents the first deep learning approach for predicting functional connectivity from regional gene expression in humans**. Prior work in this space, such as the Connectome Model and its low-rank form [1], has remained limited to bilinear and feature-based methods, with deep learning only proposed as future work. While deep learning has been extensively applied to gene expression and connectomics separately, it has not been applied to this joint prediction problem. Given the relevance of this task, we emphasized careful model benchmarking, cross-dataset validation, and custom null model construction, which are paramount to addressing what the reviewer notes is a fundamental question linking molecular biology to brain function.
> - *A key motivation for using a transformer-based architecture is its flexibility for testing targeted biological hypotheses.* SMT enables systematic integration of structured genomic priors—such as TSS-sorted sequences, positional embeddings, and contextual [CLS] tokens—following trends in models like CellSpliceNet [2]. While MLPs and CNNs can incorporate similar features, **transformers are inherently designed for processing sequential information making such implementations more direct. The reviewer suggests valuable extensions to our ablation studies, including genomic order shuffling and alternative convolutional encoders, which we explore below.
> - Feedback from several reviewers motivated an **extended analysis of Figure 9 to better understand the gene groups relevant to different functional subnetworks (full details in reviewer gLbH response)**. By summing attention weights within each subnetwork, we identified three cortex-specific gene tokens and one subcortical token that consistently received high attention. Gene Ontology analysis revealed distinct modules: **a cortical module enriched for positive regulation of dendrite extension, and a subcortical module enriched for negative regulation of neuron projection development**. This finding is notable because, while functional connectivity is often associated with metabolic or activity-driven pathways, our results suggest a stronger link to developmental wiring mechanisms that persist into adulthood aligning with findings from human developmental literature and mouse studies [3][4].
>
> ## SMT architecture
> *Effect of positional information*
> The reviewer raises an important question regarding the role of positional information in SMT performance, particularly given its biological motivation and architectural contrast with the MLP. To address this, we conducted a focused ablation experiment where we varied only the token dimension (20 vs. 60), gene order (random vs. TSS-sorted), and inclusion of ALiBi positional slopes, holding all other hyperparameters constant. Each configuration was evaluated across the same 40 train-test splits used in Table 1.
>
> **SMT**
> | Configuration | Mean r | Std Dev |
> |--------------------|--------|---------|
> | Token dim 20, random genes, no slopes       | 0.765 | 0.0291 |
> | Token dim 20, random genes, slopes          | 0.763 | 0.0277 |
> | Token dim 20, sorted genes, slopes          | 0.765 | 0.0299 |
> | **Token dim 60, random genes, no slopes**   | **0.777** | **0.0308** |
> | **Token dim 60, random genes, slopes**      | **0.781** | **0.0287** |
> | **Token dim 60, sorted genes, slopes**      | **0.783** | **0.0288** |
>
> **SMT w/ CLS**
> | Configuration | Mean r | Std Dev |
> |--------------------|--------|---------|
> | **Token dim 20, random genes, no slopes**   | **0.815** | **0.0384** |
> | **Token dim 20, random genes, slopes**      | **0.829** | **0.0277** |
> | **Token dim 20, sorted genes, slopes**      | **0.831** | **0.0276** |
> | Token dim 60, random genes, no slopes       | 0.803 | 0.0396 |
> | Token dim 60, random genes, slopes          | 0.818 | 0.0301 |
> | Token dim 60, sorted genes, slopes          | 0.818 | 0.0297 |
>
> The ablation points towards several trends. First, model performance depends on token dimensionality: the base SMT performs best with token dim 60, whereas SMT w/ CLS favors token dim 20. Second, adding ALiBi slopes generally improves performance, particularly for the SMT w/ CLS even when gene groups are randomly ordered, Third, sorting genes by TSS provides a consistent but modest performance boost across both models, suggesting that the biological sorting may be most useful for interpretability purposes. We will include these experimental results in the section A2 of the supplement.
> While the performance margins are modest, they trend consistently in the expected direction, pointing to the potential utility of modeling positional priors, something not directly possible in an MLP. Future work can extend this analysis by grouping genes based on regulatory networks.
>
> ### *Convolutions*
> The learned linear projection in our model follows the standard query-key-value (QKV) attention mechanism, where each token is linearly projected from its original dimension (20–60) into a higher-dimensional space although convolutional strategies may offer advantages for capturing local structure. This design choice aligns with the biological nature of gene expression data, which reflects aggregate signals from multiple mRNA transcripts and isoforms hybridizing to microarray probes. As for long convolutions vs transformer blocks, we opted for transformer blocks due to their sequential design features and attention head interpretability. Recent models like AlphaGenome [5] combine both strategies, and we are eager to explore this hybrid approach in future iterations of SMT. This information will be detailed in the Methods section of the revised manuscript.
>
> ### *Incorporating pretrained models*
> We appreciate the reviewer’s excellent suggestion regarding the use of pretrained gene expression models. We chose to train our model end to end primarily due to the substantial distributional shift between the AHBA bulk microarray data and the single-cell RNA-seq data typically used in large-scale pretrained models like scBERT [6]. Microarray data are normalized to a 0–1 range and often Gaussian distributed, whereas RNA-seq data are typically represented as raw or log-transformed counts, leading to different statistical properties and input dynamics. As noted in a recent benchmarking study, “the underlying data type used in the creation of the embeddings is the most critical factor influencing performance, rather than the specific algorithm or embedding dimension” [7]. That said, we strongly agree that adapting or fine-tuning pretrained models represents a highly compelling future direction.
>
> ### *Embedding analysis*
> In response to the comment on embeddings, we believe future work could apply UMAP or PHATE [8] to edge-level embeddings to potentially reveal more structure than region-level projections. We defer this to future work or the final manuscript since figures are not permitted in the rebuttal phase.
>
> ### *Loss functions*
> All models in our study were trained using mean squared error (MSE) loss, which we will explicitly clarify in the revised manuscript. During early model development, we observed stable learning dynamics using this loss function. We agree that alternative sampling strategies, such as those that ensure balanced representation of short-, medium-, and long-range connections are a valuable extension. This could be especially powerful when paired with a distributional loss function (e.g., negative log-likelihood or KLD), which would allow the model to tune to the full distribution of target values rather than point estimates alone. Our current strategy adopts a simple curriculum-like approach by exposing the model to perturbed targets early in training and gradually shifting to the true targets. However, future iterations could benefit from more targeted or difficulty-aware sampling schemes, which will be stated in the Methods section of the manuscript.
>
> ### *Gene selection criteria*
> Since our fundamental goal is to link the transcriptome and connectome at the population-level, we sought to retain genes that are stable across individuals while enabling the model to learn from the high-dimensional feature space. To achieve this, we applied a differential stability filter using the abagen toolbox, which selects genes with consistent regional expression profiles across all AHBA donors (see Eq. 3 in response to reviewer a6QV). This yielded a final set of 7380 genes, balancing dimensionality with cross-donor reliability. While this filtering may omit some potentially informative genes, it avoids injecting individual-level artifacts. We will make the specific steps clear in section A.2.
>
> References
> 1. Qiao, M. Deciphering the genetic code of neuronal type connectivity through bilinear modeling. 2024.
> 2. Afrasiyabi, A. et al. CellSpliceNet: Interpretable multimodal modeling of alternative splicing across neurons in C. elegans. 2025
> 3. Vogel, J. W. et al. Deciphering the functional specialization of whole-brain spatiomolecular gradients in the adult brain. 2024
> 4. French, L. et al. Large-scale analysis of gene expression and connectivity in the rodent brain: insights through data integration. 2011
> 5. Avsec, Ž. et al. AlphaGenome: advancing regulatory variant effect prediction with a unified DNA sequence model. bioRxiv, 2025
> 6. Yang, F. et al. scBERT as a large-scale pretrained deep language model for cell type annotation of single-cell RNA-seq data. 2022
> 7. Zhong, J., et al.. Benchmarking gene embeddings from sequence, expression, network, and text models for functional prediction tasks. 2025
> 8. Moon, K. R. et al. Visualizing structure and transitions in high-dimensional biological data. 2019

---

### Official Review · Reviewer_Jt3G · 2025-07-09

**Clarity:** 2
**Significance:** 2
**Originality:** 2
**Rating:** 4
**Confidence:** 3

**Summary:**

This paper proposes a method for predicting population functional connectivity from transcriptomic data. The method, a spatiomolecular transformer (SMT), predicts from gene expression data ordered by their transcription start site and brain regional spatial coordinates in a self-attention mechanism, the embeddings of which are then passed to an MLP for functional connectivity prediction. The method is compared against baselines and results using spatiomolecular null maps in experiments using the Allen brain atlas for the genetic data and UKBiobank and MPI-lemon for the functional connectivity data.

**Questions:**

1. Please see the questions/concerns in the weaknesses described above. I expand upon a few below.

2. I am not so sure of the validity of the conceptual idea of predicting the functional connectome from gene expression data for a small number of subjects, from a different dataset, at the population level - I welcome further discussion from the authors on this.

3. For the experiments, I would like to better understand the significance of presented results - how the small difference between the proposed method's performance and null model compared to baseline approaches might mean, variation in the runs.

**Ethical Concerns:**

["NO or VERY MINOR ethics concerns only"]

**Final Justification:**

Please see longer response to rebuttal below. In summary, I have increased my score from 2 to 4, given the authors' clarifications, the new experiments with the more challenging ROI split, and the inclusion of standard deviations in the reporting of results address many of my questions, but I still have some remaining concerns regarding novelty/effectiveness of the proposed approach compared to simpler deep learning models and the clarity of presentation of the paper itself.

**Limitations:**

yes

**Quality:**

2

**Strengths And Weaknesses:**

Strengths:
1. The motivation of the paper is to understand the link between functional connectivity and gene expression data. Such understanding would make a large impact in neuroscience.
2. The approach of tokenizing the gene expression data by TSS makes sense in terms of keeping physically contiguous data together.
3. The experimental approach includes analysis of spatiomolecular null maps to give a sense of whether the correlations learned by the model are meaningful.
4. The experiments include some generalization experiments of model performance across atlas resolutions and datasets.

Weaknesses:
1. Concern regarding generalizability of trained models:
The Allen brain atlas includes data from 6 donors - I think it is difficult to accept that a model learned from such a small number of subjects would be generalizability to the greater population. Also, the UK biobank is generally older subjects, and the MPI dataset are young adults - I am not sure it makes sense to build a model to predict between these datasets.
2. Limited technical novelty:
The novelty in this work may lie in the use of established / other methods (transformer style model, ALiBI biases, use of [CLS] token, etc) for the specific task of predicting functional connectome from gene expression data.
3. Questions/concerns regarding experimental setup:
- The experimental setup involves reserving 75% of functional connections for training, and using 25% for testing, in prediction of a population functional connectome from population gene expression data. However, one may imagine that the connections in nearby or homologous regions may have similar relationships to the genetics - therefore, there may be some inadvertent leaking of information, making the training and testing sets not independent, and thus the results may be inflated.
- The authors perform 10 random 4-fold splits for validation - I am not sure if this means 10 x 4-fold CV, or just 10 random splits where 25% was reserved for testing.
4. Questions/concerns regarding significance of results:
- The authors perform multiple runs for the experiments which is great, but then only report the means. Reporting some sense of the variance across runs would be informative in understanding of there is really a difference in results.
- While the authors do report some statistical testing results, it is for one-sided t-test, which is not best practice, unless there is a strong reason to not consider two-sided (which I am not seeing here).
- The authors run a nice experiment to compute the correlation for a null model. While most compared approaches had a null r of 0.3-0.4 range, the proposed method had a null r of 0.71, yet the results of the proposed method with the real data are at similar to slightly higher levels than competing approaches (around 0.7-0.8). Thus, to me this suggests that the proposed approach suffers from more overfitting as it was able to create a model from the perturbed input data with very high correlation compared to baselines, yet the proposed model on the real data has similar predictive strength as other models.
- The ablation experiments show very small nominal differences in performance which are likely with in standard deviation (unsure if that is what the sigmas refer to in the figure).
5. Proofreading needed:
The paper would benefit from additional proofreading to polish the language (edit for grammar, redundant words, etc). Also please check reporting of results in lines 324-326 (seems to be repeat of experiment setup but then reported different performance).

---

> ### Author Rebuttal · Authors · 2025-07-31
>
> We sincerely thank reviewer Jt3G for their thoughtful and constructive feedback. We agree with the reviewer on several key points: the conceptual validity of modeling population-level data requires careful justification, and that the experimental train/test splits must be robust to spatial leakage confounds. We address the concerns about data validity by discussing a set of recent papers and pipelines that have shown the transcriptomic stability of the AHBA dataset. Furthermore, we have performed two major new sets of experiments that demonstrate the generalization phenomena of our models despite the pervasive autocorrelation present in the datasets. First, to bolster our generalizability claims, we have replicated our cross-dataset analysis using the HCP-young-adults (YA) dataset, showing a stable adult connectome and consistent predictive performance.
>
> Second, we have implemented a more challenging spatial train-test split, which confirms that our models generalize to spatially distinct regions well above our stringent null model baseline. Together, these new results and clarifications strengthen our central claim: we have developed a novel and rigorous platform for predicting functional connectivity from gene expression using deep non-linear models that learns genuine biological signal beyond spatial autocorrelation. We address the major comments of the reviewer below in further detail and we state our plans to revise the manuscript accordingly with the findings from our new experiments.
>
> ## Conceptual validity
> We thank R1 for raising the point on the validity of using the AHBA transcriptome (from 6 subjects) to predict the connectivity strength in UKBB and MPI connectomes (two age divergent cohorts). Our work models the canonical, population-level relationship between the transcriptome and connectome, a link supported by prior work [1][2]. To ensure our data reflects this stable adult architecture, we took two key steps to ensure that the input transcriptome and output connectomes reflect stable values.
>
> ### *Stable Transcriptome:*
> We used the abagen toolbox to create a robust population-level gene expression map from the 6 AHBA donors. As the gold standard dataset in human neuroscience, ample effort has been dedicated to establishing community standard preprocessing steps [3][4]. This standard practice includes stringent processing to retain only genes with spatially consistent expression patterns across all donors based on a **differential stability threshold explained in a detailed response to a6QV**. As shown by Vogel et al. (2024) [5], the primary gene expression gradients from this 6-donor map are highly correlated (r>0.86) with those from much larger datasets (GTEx, n=121), confirming its validity.
>
> ### *Stable Connectome:*
> To address the age-mismatch concern, we have now replicated the experiment shown in Figure 7 using a new connectome derived from the HCP-YA (Human Connectome Project Young Adult, n=1065) dataset. First, we found the HCP-YA (age 22-35) and UKBB (age 45-82) connectomes to be remarkably similar (r=0.9). Crucially, we found that models trained on either cohort generalize equally well to the younger MPI-LEMON (age 20-30) cohort above null chance, despite age differences. The full results will be added to Fig. 7. This new result confirms our results are robust across cohorts of differing demographics.
>
> | Source      | MPI-183 Target True / Null r |
> |---------------|----------------|
> | UKBB   | 0.78 / 0.59        |
> | HCP-YA | 0.73 / 0.59        |
>
>
> ## Train-test split
> The reviewer raises a valid concern that the random split procedure may leak information due to spatial proximity between training and test regions. This is an excellent point. We recognize that random splits can be susceptible to spatial confounds.
>
> Our initial random-split design (10 rounds of 4-fold CV train/test split) was paired with a novel spatiomolecular null model precisely to quantify and control for this effect (described in sections 3.3. and A.3). The substantial gap between the true and null models (Table 1) provided initial evidence of learning beyond spatial proximity. To provide even stronger evidence, we have now implemented a more challenging spatial split. We do this by selecting a source region and assign the edges from the 25% nearest regions to the test set, with the rest used for training [6]. This is done across 4 folds to cover the whole brain, and repeated 10 times, yielding 40 splits, evaluating generalization to spatially distinct regions. All models are evaluated for the same 40 splits just as was done in the random setting. The results, summarized below, are highly informative:
>
> | Model           | Random Split (True / Null r)        | Spatial Split (True / Null r)       |
> |-----------------|--------------------------------------|--------------------------------------|
> | **SMT w/ [CLS]**   | 0.817 ± 0.028 / 0.713 ± 0.047        | 0.743 ± 0.114 / 0.516 ± 0.111        |
> | **MLP w/ coords** | 0.808 ± 0.031 / 0.466 ± 0.078        | 0.741 ± 0.126 / 0.476 ± 0.098        |
>
> These new results show that while the null performance of SMT w/ CLS drops by ~0.20 in the harder split (confirming that spatial information is less useful in this regime), the true performance only drops by ~0.08. **This doubles the performance gap between the true and null models**, providing compelling evidence that SMT learns a genuine, spatially generalizable transcriptomic-connectomic mapping. We will add this new train/test split routine to the revised manuscript in a dedicated table.
>
> ## Experimental details
> *Standard deviations & statistical testing*.
> To address reviewer concerns, we will add standard deviations to all our reported results including the ablation experiments. They were initially omitted in the tables due to formatting constraints. They also note our use of a one-sided t-test, which we have replaced with two-sided t-tests. The two sided tests confirm the same patterns that the SMT w/ CLS performs significantly better on long-r (p=0.014) and geodesic distance (p=0.0004) than MLP w/ coords.
>
> ## Novelty
> Lastly, we acknowledge the reviewer’s observation that some SMT components are adapted from existing techniques. However, we highlight several aspects of novelty in both modeling and experimental design:
> - **First deep learning approach for transcriptome-to-connectome mapping in humans**: Prior work in transcriptome-connectome prediction focuses only on feature-based or bilinear methods citing deep learning approaches as future work [7]; we are the first to apply both MLP and attention-based models in this setting.
> - **Rigorous benchmarking framework**: We systematically compare feature-based, bilinear, and deep models in a unified setting using a spatiomolecular null to isolate biological signal from spatial autocorrelation effects.
> - **Consistent non-linear performance gains**: Nonlinear models, including SMT, outperform existing linear baselines under several split settings. Full spatial split results across all models will be reported in a dedicated table of the final manuscript.
> - **SMT modularity enables testing biologically informed priors**: The transformer architecture allows structured integration of genomic and spatial priors through positional embeddings and contextual tokens, which we find improve performance. This modularity provides a natural foundation for future extensions using [CLS]-like tokens to incorporate additional biological or demographic context.
> - **Interpretability via attention weights**: Building on Supplement Figure 9 (see response to reviewer gLbH), we identified cortical and subcortical specific tokens enriched for gene modules related to dendrite extension and neuronal projection.
> - **Future potential**: While SMT is tested here in a small-scale setting, transformer architectures are particularly well-suited to more data-rich regimes. Having demonstrated its utility and performance above baselines, SMT can be scaled to individualized modeling with larger transcriptomic datasets and extended through integration with pretrained genomic models such as scBERT [8] or Geneformer [9].
>
> References
> 1. Vértes, P. E. et al. Gene transcription profiles associated with inter-modular hubs and connection distance in human functional magnetic resonance imaging networks. 2016
> 2. Anderson, K. M. et al. Gene expression links functional networks across cortex and striatum. 2018
> 3. Arnatkeviciūtė, A. et al. A practical guide to linking brain-wide gene expression and neuroimaging data. 2019.
> 4. Markello, R. D. et al. Standardizing workflows in imaging transcriptomics with the abagen toolbox. 2025
> 5. Vogel, J. W. et al. Deciphering the functional specialization of whole-brain spatiomolecular gradients in the adult brain. 2024
> 6. Hansen, J. Y. Mapping gene transcription and neurocognition across human neocortex. 2021
> 7. Qiao, M. Deciphering the genetic code of neuronal type connectivity through bilinear modeling. 2024
> 8. Yang, F. et al. scBERT as a large-scale pretrained deep language model for cell type annotation of single-cell RNA-seq data. 2022
> 9. Cui, Z. et al. Geneformer: Learned gene compression using transformer-based context modeling. 2023

---

> > ### Author Response · Authors · 2025-08-07
> >
> > Dear reviewer Jt3G, we are writing to follow up on our recent rebuttal. We thank you for your initial comments and helpful suggestions and have worked diligently to address the critical concerns you raised.
> >
> > In our detailed rebuttal, we presented two major new experiments that directly address your primary concerns about data validity and spatial confounds.
> > - We validated our model on the HCP-YA dataset, confirming that our approach is robust across age-divergent cohorts.
> > - We also implemented a much more conservative spatial train-test split, demonstrating that our models learn genuine biological signal that generalizes to unseen regions, well beyond the effects of spatial proximity.
> >
> > As a brief summary of those key results, the performance in the spatial split (table below) confirms that non-linear models significantly outperform strong linear baselines, and crucially, the performance gap between our true and null models widens considerably in this harder setting, providing strong evidence against overfitting. See the summary of this table below.
> > | **Model**                  | **True $r$**      | **Null $r$**      |
> > |----------------------------|--------------------|--------------------|
> > | SMT w/ [CLS]              | 0.743 ± 0.114      | 0.516 ± 0.111      |
> > | MLP w/ coords | 0.741 ± 0.126      | 0.476 ± 0.098      |
> > | SMT                       | 0.734 ± 0.130      | 0.301 ± 0.123      |
> > | MLP           | 0.726 ± 0.129      | 0.280 ± 0.116      |
> > | Low-Rank Bilinear     | 0.669 ± 0.131      | 0.336 ± 0.131      |
> >
> > We believe these new results and our detailed responses substantially strengthen the paper. If you find that these additions have adequately addressed your concerns, we would be very grateful if you would consider updating your review and score. We look forward to your response and are open to any further discourse or clarifications.

---

> ### Comment · Reviewer_Jt3G · 2025-08-07
> **Response to rebuttal**
>
> Thanks to the authors for this great rebuttal (and also the responses to other reviewers comments, which was very helpful). The rebuttal addresses many of my concerns. I greatly appreciate the new experiments and analysis; in particular, I find the experiment with the more difficult test/train split of the ROIs to be very informative as well as the inclusion of the standard deviations in the CV experiments, as we can now much better assess the difference between the true and null results. I understand that a main motivation of the paper is the application of the nonlinear deep learning approaches for the gene expression to functional connectome prediction task - but the proposed SMT still does not to me show convincing improvement over the simpler MLP model, and the technical approach itself is still largely derived from existing methods. I understand the authors points regarding interpretability, but it is unclear if using some standard explanation approaches for the MLP model may also give meaningful interpretation. Also, while the rebuttals are presented with great clarity, the original paper itself I found had presentation issues, and it is hard to know whether they will be resolved upon final submission. Given the additional nice results and explanations in the rebuttal but some remaining concerns, I choose to increase my rating from 2 to 4.

---

> > ### Author Response · Authors · 2025-08-07
> >
> > We thank the reviewer for their thoughtful follow-up and for recognizing the value of our new experiments and clarifications. These additions will be integrated into the manuscript to better support the core contributions of our work: we introduce the first deep learning framework for predicting functional connectivity from regional gene expression in humans and demonstrate that non-linear models (SMT, MLP) consistently outperform linear baselines and exceed chance-level performance, using a novel spatiomolecular null and multiple train-test split settings. The manuscript has been revised to more clearly reflect these contributions, guided by the helpful suggestions from all reviewers. We are grateful for the reviewer's engagement and are pleased that the improvements addressed their main concerns.

---

### Note · Authors · 2025-08-15

We sincerely thank the reviewers for their insightful comments. Their feedback directly motivated us to conduct two major new experiments and revise our manuscript, which we believe have made the paper much stronger.

First, to address the critical concern of spatial information leakage in our train/test splits, we implemented a challenging new spatial split, where test connections are from regions spatially distinct from the training set. This analysis provides compelling evidence for genuine biological learning: while our stringent null model's performance dropped substantially (from r=0.71 to 0.52), our SMT model's true performance only dipped slightly (from r=0.82 to 0.74). This doubles the performance gap over the null baseline, strongly refuting the hypothesis that our results are inflated by spatial autocorrelation.

Second, to address concerns about the conceptual validity of using the 6-donor AHBA dataset and generalizing across age-divergent cohorts, we replicated our cross-dataset analysis using the HCP-Young Adult connectome. We found the young adult (HCP) and older adult (UKBB) connectomes are remarkably similar (r=0.9), and models trained on either generalize equally well to the MPI-LEMON dataset. This confirms our approach successfully models a stable, canonical brain architecture, robust to demographic differences in the source data.

We also clarify for Reviewer NCYX that our work is the first to apply deep learning to predict the human connectome from the transcriptome. Our contribution lies in establishing a rigorous evaluative framework with a novel spatiomolecular null model and demonstrating that non-linear models consistently outperform prior linear methods. The transformer's modularity is a key feature, enabling the structured integration of biological priors. As the first work to introduce this fundamental neuroscientific problem in a deep learning context, we believe our paper will stimulate interesting discussion at NeurIPS.

---

### Decision · Program_Chairs · 2025-09-17

**Decision:**

Accept (poster)

**Comment:**

Summary
This paper introduces the Spatiomolecular Transformer (SMT), a transformer-based framework for predicting functional brain connectivity from population-level regional gene expression. Intuitively, the architecture has two stages: first, an encoder builds region-level representations from transcriptomic data (Allen Human Brain Atlas), grouping genes into tokens ordered by transcription start site and adding a region CLS token processed by a transformer to output a region embedding; second, a decoder MLP uses pairs of these region embeddings to predict functional connectivity. The transcriptomic atlas is constant (6 AHBA donors), while target functional connectivities are varied during training across subjects (these are from UKBB, MPI-LEMON, HCP-YA). This pairing allows the model to generalize by exposing the fixed transcriptome to multiple subject-level connectomes (also functioning as label augmentation). A novel spatiomolecular null baseline is introduced to control for spatial autocorrelation.

Strengths
- Tackles a fundamental and timely neuroscience problem with careful experimental design.
- Rigorous evaluation across datasets, parcellations, and train/test split strategies, including stringent spatial splits and null baselines.
- Demonstrates that nonlinear models (MLP, SMT) consistently outperform linear and bilinear methods, with SMT showing additional strengths for long-range/geodesic connections.

Weaknesses
- Gains of SMT over simpler MLP baselines are modest, raising questions about the necessity of the transformer.
- Gene tokenization by TSS is based on a rough biological prior (genomic distance).

Decision Rationale
The consensus among reviewers is that this is a technically sound and carefully executed paper whose main strength lies in the modeling for linking transcriptomic and connectomic data, with thoughtful baselines and evaluations. The rebuttal added substantial new experiments (spatial splits, HCP-YA replication) that directly addressed key technical concerns and convinced one initially negative reviewer to raise their score, leading to a positive consensus. While the incremental gains of SMT over MLP remain debated, the reviewers agreed that the overall contribution is meaningful: it demonstrates the utility of nonlinear models in this domain while maintaining interpretability despite added complexity. I recommend poster here given the specialized scope of the application.

Reviewer Discussion
- All reviewers engaged constructively; the authors’ new experiments on spatial splits and HCP replication were well received.
- Reviewer Jt3G raised their score significantly after the rebuttal, shifting the consensus toward acceptance.